# Towards Long-Horizon Interpretability:
# Efficient and Faithful Multi-Token Attribution for Reasoning LLMs

**Wenbo Pan** [* 1]  **Zhichao Liu** [* 2]  **Xianlong Wang** [1]  **Haining Yu** [2]  **Xiaohua Jia** [1]

## Abstract

Token attribution methods provide intuitive explanations for language model outputs by identifying causally important input tokens. However, as modern LLMs increasingly rely on extended reasoning chains, existing schemes face two critical challenges: (1) efficiency bottleneck, where attributing a target span of $M$ tokens within a context of length $N$ requires $\mathcal{O}(M \cdot N)$ operations, making long-context attribution prohibitively slow; and (2) faithfulness drop, where intermediate reasoning tokens absorb attribution mass, preventing importance from propagating back to the original input. To address these, we introduce FLASHTRACE, an efficient multi-token attribution method that employs span-wise aggregation to compute attribution over *multi-token targets in a single pass*, while maintaining faithfulness. Moreover, we design a recursive attribution mechanism that traces importance through intermediate reasoning chains back to source inputs. Extensive experiments on long-context retrieval (RULER) and multi-step reasoning (MATH, MorehopQA) tasks demonstrate that FLASHTRACE achieves over 130× speedup over existing baselines while maintaining superior faithfulness. We further analyze the dynamics of recursive attribution, showing that even a single recursive hop improves faithfulness by tracing importance through the reasoning chain. Our code is available at https://github.com/wbopan/flashtrace.

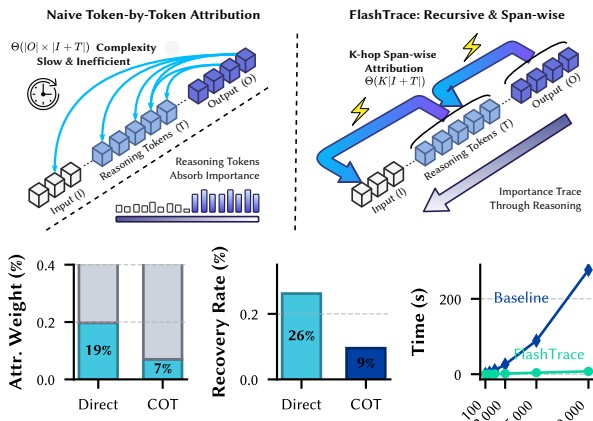

*Figure 1.* Motivation for FLASHTRACE. **Top:** Naive token-by-token attribution requires expensive per-token computation, while FLASHTRACE performs efficient span-wise recursive attribution. **Bottom:** (a) With extended reasoning, attribution weight on reasoning tokens increases significantly (from approximately 80% to over 90%); (b) This causes recovery rate of ground-truth input tokens to drop substantially (from 26% to below 10%); (c) Naive multi-hop attribution scales poorly with reasoning length, while FLASHTRACE remains efficient even for 10K tokens.

## 1. Introduction

While more and more high-stakes decisions are made by Large Language Model (LLM) agents (Novikov et al., 2025), interpreting their outputs becomes increasingly important. Token attribution methods offer an intuitive explanation approach (Achtibat et al., 2024; Ferrando & Voita, 2024). Given a specific output to explain, these methods calculate an importance distribution over all input tokens to identify those causally responsible for the generation. This provides principled leverage to both understand LLM behaviors and optimize context.

However, as recent reasoning and agentic LLMs generate long sequences of tokens to solve tasks (Chen et al., 2025), existing attribution methods scale poorly to *multi-token attribution*. Since existing approaches target the attribution of a single token, they suffer from being applied to multiple LLM-generated tokens, leading to two key limitations: **(1) Efficiency Bottleneck**. Explaining a sequence naively requires iterating over each target token, leading to prohibitive costs. For instance, classic methods like *Integrated Gradi-*

---

[*]Equal contribution  [1]Department of Computer Science, City University of Hong Kong, Hong Kong SAR, China [2]Harbin Institute of Technology, Harbin, China. Correspondence to: Haining Yu <yuhaining@hit.edu.cn>.

*Proceedings of the 43rd International Conference on Machine Learning*, Seoul, South Korea. PMLR 306, 2026. Copyright 2026 by the author(s).

*ents (IG)* (Vafa et al., 2021) can require over 10 hours to explain a single 5k-token generation, making them impractical for production use. **(2) Faithfulness Drop**. In multi-token generation, the LLM no longer produces decisions based directly on the input; instead, it answers based on intermediate results within a reasoning chain. Existing attribution methods only handle direct input-output dependencies, causing attribution mass (i.e., the total importance score) to concentrate on the reasoning tokens. They fail to trace the dependency from the reasoning chain back to the input, resulting in an inability to locate high-importance input tokens.

In this work, we introduce FLASHTRACE, an efficient multi-token attribution method optimized for the reasoning and agentic workflows of LLMs, where we explain a contiguous span of model-generated tokens rather than single tokens. Instead of iterating through the target sequence token-by-token, FLASHTRACE computes the contribution of context tokens to a multi-token target in a single pass per hop. To address the issue where reasoning tokens absorb attribution mass, we design a *Recursive Attribution* mechanism. This process handles the internal causal dependencies of multi-token generation within a small number of recursive hops. As a result, our method accelerates the process from hours to seconds and enables efficient multi-token attribution.

Through extensive experiments, we show that FLASH-TRACE surpasses baselines in both faithfulness and efficiency for long-context and reasoning tasks, particularly in scenarios requiring complex reasoning like multi-hop QA. Furthermore, we analyze how the attribution results from the model's final output propagate along the reasoning chain to the input during recursive attribution. We show that the performance gains from this process are consistent across different models and data distributions, highlighting the importance of attributing *through* the reasoning chain. Our core contributions are summarized as follows:

◇ We formalize the *multi-token attribution* problem for reasoning LLMs and identify two key challenges: the efficiency bottleneck and faithfulness degradation caused by intermediate reasoning tokens.

◇ We propose FLASHTRACE, a novel attribution method that addresses these challenges via span-wise aggregation and recursive attribution, enabling efficient and faithful multi-token attribution.

◇ We conduct extensive experiments demonstrating that FLASHTRACE achieves significant speedup while maintaining superior faithfulness across different task domains and model architectures.

◇ We analyze the dynamics of recursive attribution, revealing how attribution distributions shift across hops and quantifying their effect on final attribution quality.

**Conflict of Interest Disclosure.** The authors declare no financial conflicts of interest. All language models evaluated in this work are publicly available and were not developed by any organization affiliated with the authors.

## 2. Related Work

**Token Attribution for Transformers.** Token attribution methods assign importance scores to input tokens to explain model predictions. Existing approaches include perturbation-based methods that measure output changes when tokens are removed (Liu et al., 2024; Cífka & Liutkus, 2023; Zhao & Shan, 2024), gradient-based methods such as Integrated Gradients (Sundararajan et al., 2017; Vafa et al., 2021), and attention-based methods that combine attention weights with gradient signals (Chen et al., 2022). More recently, methods such as IFR (Ferrando & Voita, 2024) and AttnLRP (Achtibat et al., 2024) achieve attribution by analyzing relevance propagation through transformer components. While effective for single-token targets, these methods scale poorly to multi-token outputs and fail to trace importance through chains of intermediate reasoning. Concurrent work CAGE (Walker & Ewetz, 2025) addresses multi-hop attribution by recursively tracing importance through reasoning chains; however, it operates at sentence-level granularity and incurs high computational cost, limiting its applicability to long-context scenarios. A complementary line of interpretability research analyzes internal feature circuits rather than input-token importance, including transcoders (Dunefsky et al., 2024) and circuit-tracer (Hanna et al., 2025); we discuss how these methods relate to FLASHTRACE as additional related work in Appendix J.

**Reasoning LLM and Long-Context Agent.** Recent advances in LLMs have popularized chain-of-thought reasoning (Wei et al., 2022) and tool-calling paradigms (Yao et al., 2023; Schick et al., 2023), where models interleave extended reasoning with external interactions. Recent models such as OpenAI o1 (Jaech et al., 2024) and DeepSeek-R1 (DeepSeek-AI, 2025) produce thousands of reasoning tokens through reinforcement learning, creating long chains where attribution must traverse multiple hops to identify causally relevant inputs. These developments demand attribution methods capable of handling large-scale contexts with complex token dependencies.

## 3. Multi-Token Attribution

### 3.1. Background

**LLM Reasoning.** Unlike early language models that generated answers immediately upon receiving input, modern LLMs often engage in extended reasoning processes. Before providing a final answer, these models generate intermediate

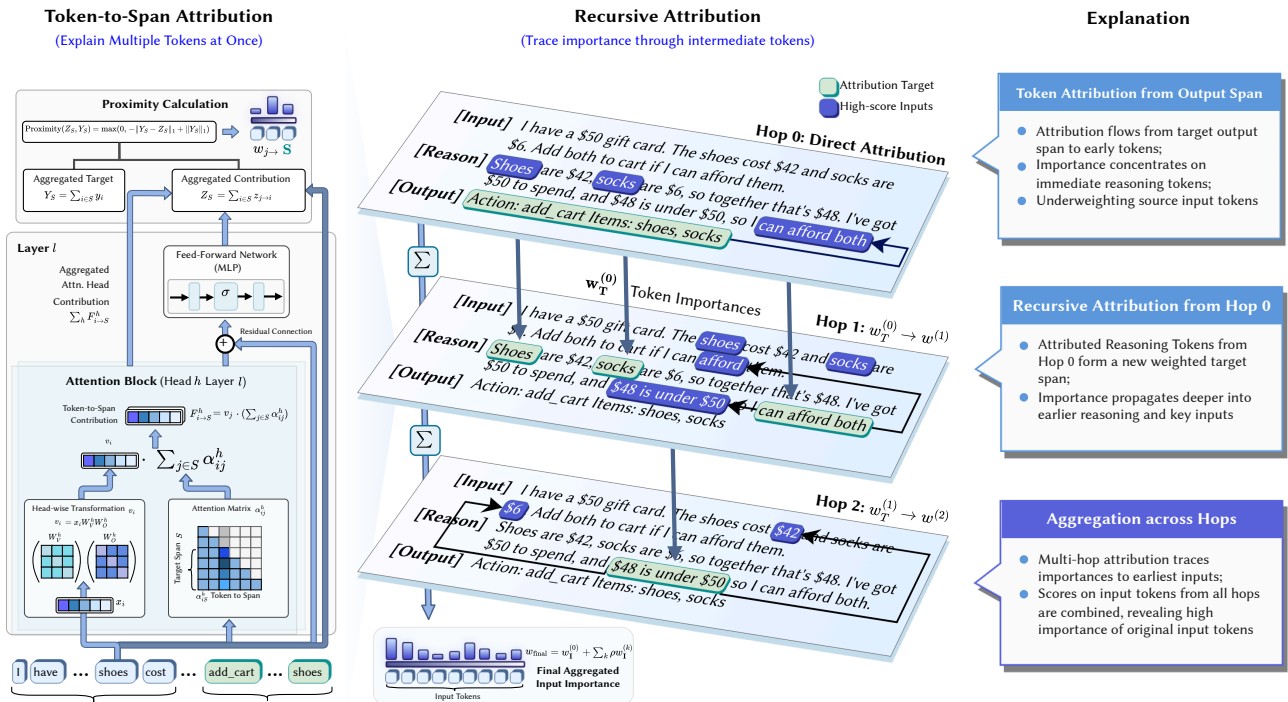

*Figure 2.* Overview of FLASHTRACE span-wise attribution. The method pre-aggregates causal attention and residual contributions within target spans at each layer, enabling efficient multi-hop attribution through chains of intermediate reasoning.

tokens to derive proofs, explore solution spaces, or reflect on prior context. These reasoning tokens can constitute the vast majority of a model's generation, sometimes spanning thousands of positions. Formally, we decompose a full model context into three segments: input tokens $\mathbf{I}$, intermediate reasoning tokens $\mathbf{T}$, and final output tokens $\mathbf{O}$. While both $\mathbf{T}$ and $\mathbf{O}$ are generated by the model, they serve distinct roles: $\mathbf{T}$ facilitates the computation required to produce $\mathbf{O}$. The full context is $\mathbf{S} = \mathbf{I} \circ \mathbf{T} \circ \mathbf{O}$, concatenation of 3 segments.

**Token-Level Attribution.** Token-level attribution serves as a tool for mechanistic interpretation by assigning importance scores to context tokens based on their causal influence on generated outputs. The primary goal is to identify which specific segment of the context explains a model's output. While attribution methods have been successfully validated on atomic, synthetic tasks like Indirect Object Identification (Wang et al., 2023), their application to complex, multi-token generations remains underexplored. In the following section, we demonstrate that the presence of intermediate reasoning tokens $\mathbf{T}$ introduces significant challenges to both the **faithfulness** and **efficiency** of standard attribution techniques.

### 3.2. How Reasoning Tokens Impact Attribution

Before introducing our method, we investigate a fundamental question: *Can existing token-level attribution methods effectively trace importance back to the original input when*

*the model produces a chain of intermediate reasoning?*

**Setup.** We utilize the multi-hop retrieval task from RULER (Hsieh et al., 2024), which requires the model to locate multiple key entities scattered throughout a long context. We evaluate Qwen-3 8B Instruct (Yang et al., 2025) in two configurations: *direct answering*, where the model outputs the answer directly, and *chain-of-thought* (CoT), where the model explicitly generates reasoning steps before answering. In both settings, the model answers correctly. We then apply AttnLRP (Achtibat et al., 2024) to attribute the final answer tokens $\mathbf{O}$ back to the preceding context.

**Finding 1: Reasoning tokens absorb the majority of attribution mass.** Figure 1(a) illustrates that reasoning tokens capture a substantial portion of the total importance score. As the length of the reasoning chain increases, the fraction of importance assigned to $\mathbf{T}$ grows proportionally, leaving little mass for the original input $\mathbf{I}$. This concentration on nearby tokens is structurally expected: in an auto-regressive model, each reasoning token directly conditions the generation of the next. However, this behavior is problematic for long-context interpretability. For example, when attributing a mathematical proof or code generation, we typically want to identify the original problem statement or constraints in $\mathbf{I}$ that dictated the result, rather than the immediately preceding derivation step in $\mathbf{T}$.

**Finding 2: Reasoning chains degrade attribution quality**

**on inputs.** To quantify attribution quality, we define the *recovery rate*: the fraction of ground-truth key input tokens that appear in the top 10% of attributed tokens (excluding model-generated tokens from the ranking). Figure 1(b) compares the recovery rate between direct answering and CoT. When the model employs reasoning, the attribution method fails to look past the intermediate steps, resulting in significantly lower recovery of the ground-truth inputs. This suggests that standard token-level attribution captures direct, local causal effects but fails to propagate importance *through* the reasoning chain back to the source inputs.

**Limitations of Multi-hop Attribution.** A theoretical solution is to perform attribution recursively: first attribute the output $\mathbf{O}$ to the reasoning tokens $\mathbf{T}$, and then attribute the important reasoning tokens back to the input $\mathbf{I}$. While conceptually sound, this multi-hop approach is computationally prohibitive. Standard attribution methods calculate the influence of all context tokens on a *single* target token. To attribute a reasoning span of length $M = |\mathbf{T}|$, one must run the attribution algorithm $M$ separate times, once for each token in the span. This increases the time complexity from $\mathcal{O}(N)$ to $\mathcal{O}(M \cdot N)$, where $N = |\mathbf{S}|$ is the total context length. As shown in Figure 1(c), for reasoning chains of just a few hundred tokens, this naive multi-hop attribution takes hours on a single GPU, making it impractical for long-context agents. For complete experimental details and baseline implementation, see Appendix I.

# 4. FLASHTRACE

We introduce FLASHTRACE, a component-level attribution method designed for efficient attribution over multi-token spans in long-context LLM outputs. As illustrated in Figure 2, FLASHTRACE introduces two key innovations based on proximity functions from Ferrando et al. (2022) to overcome the efficiency bottlenecks described above:

1. **Span-wise Aggregation:** Unlike conventional methods that process a single target token at a time, FLASHTRACE computes the importance of all heads, residual connections, and MLPs with respect to a **multi-token target span** in a single pass, computing attribution for the entire target span in a single pass.

2. **Recursive Attribution:** To address the information absorption by reasoning tokens, FLASHTRACE performs recursive attribution steps. It first identifies important reasoning spans from prior attribution steps and then launches new attribution passes targeting the weighted aggregation of those reasoning tokens.

By combining these techniques, FLASHTRACE effectively traces influence from output tokens $\mathbf{O}$, through intermediate reasoning $\mathbf{T}$, back to the environment inputs $\mathbf{I}$. Notably, it achieves this with a computational cost comparable to a single forward pass per hop, enabling scalable interpretability for long-context workflows.

## 4.1. Preliminaries

FLASHTRACE builds upon the theoretical framework of ALTI (Aggregation of Layer-wise Token-to-Token Interactions) and IFR (Information Flow Route) (Ferrando et al., 2022; Ferrando & Voita, 2024). This framework uses vector proximity to estimate importance scores across Transformer layers and token positions. To trace information flow, we first decompose the operations within a Transformer layer.

A Transformer layer consists of three mechanisms that move information between tokens: the multi-head attention mechanism, the feed-forward network (FFN), and the residual connection. For a given token at position $i$, the output representation $\mathbf{y}_i$ at a specific layer is the sum of the residuals, the outputs from all attention heads, and the FFN output:

$$\mathbf{y}_i = \mathbf{x}_i^{res} + \sum_{h=1}^{H} \mathbf{h}_i^h + \mathbf{m}_i, \quad (1)$$

where $\mathbf{x}_i^{res}$ is the input from the residual stream, $\mathbf{h}_i^h$ is the output of attention head $h$, and $\mathbf{m}_i$ is the output of the FFN.

To perform token-level attribution, we must further decompose the attention output. The output of an attention head $\mathbf{h}$ at position $i$ is a weighted sum of transformed vectors from all input tokens $j$. We denote the transformation function for a single input token as $\mathbf{f}_{j \to i}$:

$$\mathbf{y}_i^{attn} = \sum_{j=1}^{|\mathbf{S}|} \mathbf{f}_{j \to i}(\mathbf{x}_j) + \mathbf{b}. \quad (2)$$

Here, $\mathbf{f}_{j \to i}$ represents the contribution of input token $j$ to the target token $i$. This contribution is computed by projecting the input $\mathbf{x}_j$ through the Value and Output matrices, scaled by the attention weight $\alpha_{i,j}$:

$$\mathbf{f}_{j \to i}(\mathbf{x}_j) = \alpha_{i,j}^h \cdot \underbrace{(\mathbf{x}_j W_V^h W_O^h)}_{\text{Transformed Vector } \mathbf{v}_j}. \quad (3)$$

Ideally, we want to measure how important a specific input contribution $\mathbf{z}$ (such as $\mathbf{f}_{j \to i}$) is to the resulting representation $\mathbf{y}$. Following ALTI, we use a proximity metric based on the L1 norm, which is more robust in the high-dimensional, anisotropic vector space of Transformers:

$$\text{Proximity}(\mathbf{z}, \mathbf{y}) = \max(0, -\|\mathbf{y} - \mathbf{z}\|_1 + \|\mathbf{y}\|_1). \quad (4)$$

Intuitively, this metric measures how much the magnitude of the target vector $\mathbf{y}$ would decrease if the contribution $\mathbf{z}$ were removed. By recursively applying this metric through the network components, we can estimate the importance of input tokens relative to the final output representation. We emphasize that this proximity-based score quantifies the *informational contribution* of a source token, that is, how much of its representation propagates into the target span, rather than a direct causal effect; our perturbation-based faithfulness evaluation (Section 5) then verifies that these contribution scores reliably predict causal importance.

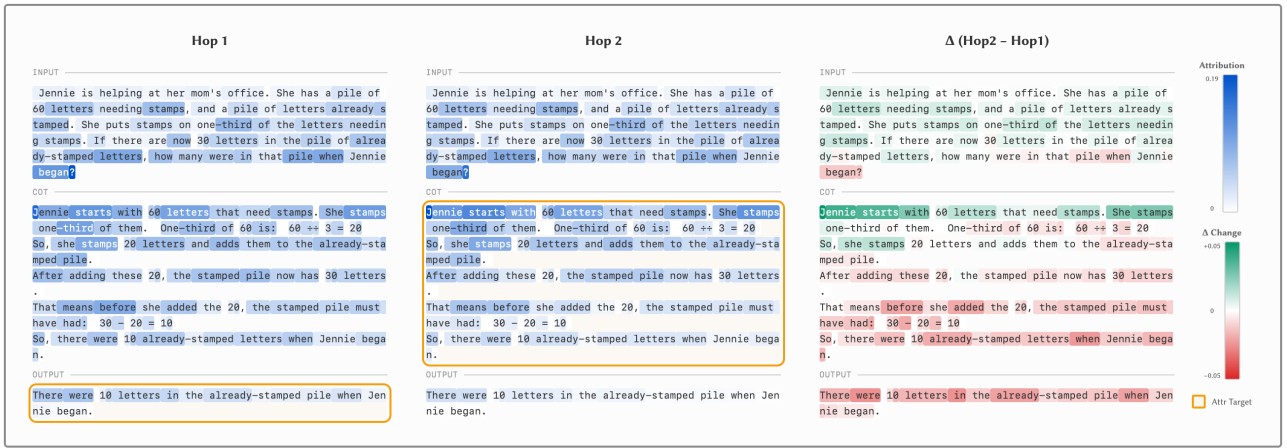

*Figure 3.* **Visualization of recursive attribution across hops.** *Hop 1*: Initial attribution concentrates on reasoning tokens nearest to the output. *Hop 2*: Attribution shifts toward earlier reasoning tokens and input context. Δ *(Hop2 - Hop1)*: The difference map shows how importance propagates backward: green regions indicate tokens gaining attribution in the second hop (typically input tokens), while red regions show tokens losing attribution (typically intermediate reasoning tokens).

## 4.2. Span-wise Aggregation

In many analysis scenarios, such as attributing a long reasoning chain or a complete tool-use sequence, we need to calculate the importance of input tokens with respect to a *set* of output tokens, rather than a single token.

A naive approach would loop over every token in the target output span. For a target span of length $M$, we would treat each token as an attribution target $\mathbf{y}_i$, compute the importance scores for the entire context of length $N$, and finally average the results. This results in a computational complexity of $\mathcal{O}(M \cdot N)$. For long reasoning chains where both $M$ and $N$ are large, this approach becomes prohibitively slow.

To overcome this bottleneck, FLASHTRACE introduces a layer-wise span-targeted aggregation objective. Instead of calculating proximity for individual tokens, we define a *target span* $S$ and compute proximity with respect to the aggregated vector of this group. We define the aggregated target $\mathbf{Y}_S$ as the sum of representations in the span:

$$\mathbf{Y}_S = \sum_{i \in S} \mathbf{y}_i. \tag{5}$$

Similarly, we define the aggregated contribution $\mathbf{Z}_S$ from a specific source component (e.g., input token $j$) as the sum of its contributions to all tokens in $S$:

$$\mathbf{Z}_S = \sum_{i \in S} \mathbf{z}_{j \rightarrow i}. \tag{6}$$

Using these definitions, we apply the same proximity metric to the group vectors:

$$\text{Prox.}(\mathbf{Z}_S, \mathbf{Y}_S) = \max(0, -\|\mathbf{Y}_S - \mathbf{Z}_S\|_1 + \|\mathbf{Y}_S\|_1). \tag{7}$$

Crucially, we can compute the aggregated contribution $\mathbf{Z}_S$ efficiently by exploiting the linearity of the attention mechanism. Recall that the contribution from token $j$ to token

$i$ is $\alpha^h_{i,j} \cdot \mathbf{v}_j$. The transformed vector $\mathbf{v}_j$ depends only on the source token $j$ and is independent of the target position $i$. Therefore, when summing over the target span $S$, we can factor out $\mathbf{v}_j$:

$$\mathbf{F}_{j \rightarrow S} = \sum_{i \in S} \left( \alpha^h_{i,j} \cdot \mathbf{v}_j \right) = \mathbf{v}_j \cdot \left( \sum_{i \in S} \alpha^h_{i,j} \right). \tag{8}$$

This algebraic reordering allows us to pre-calculate the sum of attention weights $\sum_{i \in S} \alpha^h_{i,j}$ for the target span. Consequently, we only need to compute the expensive vector transformation $\mathbf{v}_j$ once per source token, regardless of the target span length. By aggregating the residual connections in a similar manner, simply summing the residual vectors $\mathbf{x}^{res}_i$ within the span $S$, we reduce the complexity of attributing a multi-token span from $\mathcal{O}(M \cdot N)$ to $\mathcal{O}(N)$, where $M$ is the target span length and $N$ is the context length. This optimization enables FLASHTRACE to scale linearly with context length, making multi-token attribution feasible for long contexts. We provide detailed algorithmic steps in Appendix A and a detailed complexity analysis in Appendix B.

## 4.3. Recursive Attribution

As discussed in Section 3.2, while Span-wise Aggregation solves the efficiency bottleneck of multi-token targets, it does not address the "information absorption" problem. When reasoning tokens $\mathbf{T}$ are present, they tend to absorb the majority of the attribution mass, preventing importance scores from propagating back to the original input $\mathbf{I}$ that dictated the reasoning process. FLASHTRACE resolves this by performing attribution recursively. We treat the importance scores assigned to reasoning tokens in one step as weights for the target span in the next step. This allows us to trace the information flow from the final output, through the intermediate reasoning chain, back to the source input.

**First Attribution Hop.** We begin by performing a standard attribution pass on the model's final output tokens $\mathbf{O}$, where the target span length is $M = |\mathbf{O}|$. Using the span-wise aggregation method described in Section 4.2, we compute the attribution distribution $\mathbf{w}^{(0)}$ over the entire context. We can partition this distribution into two parts: the scores assigned to the original input tokens, denoted as $\mathbf{w}_{\mathbf{I}}^{(0)}$, and the scores assigned to the intermediate reasoning tokens, denoted as $\mathbf{w}_{\mathbf{T}}^{(0)}$. Typically, the total mass on reasoning tokens ($\sum \mathbf{w}_{\mathbf{T}}^{(0)}$) is high, indicating that the output is directly determined by the immediate reasoning steps. Our goal is to determine which earlier contexts dictated these specific, high-importance reasoning tokens.

**Recursive Attribution Hop.** To trace the origin of the reasoning steps, we form a new attribution target with $M = |\mathbf{T}|$. The reasoning tokens with high importance scores in the previous hop effectively become the "target" for the next hop. Specifically, we define a *weighted* target span using the reasoning tokens $\mathbf{T}$, where the weight for each token is its importance score from the previous step.

The span-wise aggregation defined in Eq. (5) naturally extends to weighted spans. That is, the aggregated target representation $\mathbf{Y}^{(1)}$ becomes a weighted sum:

$$\mathbf{Y}^{(1)} = \sum_{j \in \mathbf{T}} w_j^{(0)} \cdot \mathbf{y}_j. \qquad (9)$$

Then the aggregated contribution $\mathbf{Z}^{(1)}$ from a source token $k$ is also a weighted sum of its individual contributions:

$$\mathbf{Z}^{(1)} = \sum_{j \in \mathbf{T}} w_j^{(0)} \cdot \mathbf{z}_{k \rightarrow j}. \qquad (10)$$

We then apply the same proximity-based importance calculation using $\mathbf{Y}^{(1)}$ and $\mathbf{Z}^{(1)}$. This yields a new attribution distribution, $\mathbf{w}^{(1)}$, which represents the importance of context tokens with respect to the *relevant parts* of the reasoning chain. Intuitively, this step identifies which source tokens provided the information required to generate the critical reasoning steps found in the first hop.

It is worth noting that the efficiency benefit of span-wise aggregation is preserved in this weighted setting. The factorization in Eq. (8) simply becomes $\mathbf{v}_k \cdot (\sum_{j \in \mathbf{T}} w_j^{(0)} \alpha_{j,k}^h)$. Since the attention weights $\alpha$ and importance weights $w^{(0)}$ are scalars, the expensive vector transformation $\mathbf{v}_k$ is still computed only once per source token.

**Subsequent Hops and Aggregation.** We can repeat this recursive process for $K$ hops. In each hop $k$, we obtain a new distribution of importance scores for the input tokens, $\mathbf{w}_{\mathbf{I}}^{(k)}$, and a new distribution on the reasoning tokens, $\mathbf{w}_{\mathbf{T}}^{(k)}$, which serves as the weights for hop $k + 1$.

To derive a single, comprehensive attribution distribution, we aggregate the scores from all hops. We interpret the attri-

bution process as a flow of probability mass. In each hop, a portion of the importance flows into the input tokens, while the rest remains in the reasoning tokens to be explained by the next hop. Let $\rho_k$ be the fraction of importance assigned to the reasoning tokens in hop $k$:

$$\rho_k = \sum_{t \in \mathbf{T}} w_t^{(k)}. \qquad (11)$$

The final aggregated importance for the input tokens, $\mathbf{w}_{final}$, is the sum of the input importance from the first hop plus the input importance from subsequent hops, scaled by the residual mass carried over from previous steps:

$$\mathbf{w}_{final} = \mathbf{w}_{\mathbf{I}}^{(0)} + \sum_{k=1}^{K} \left( \prod_{j=0}^{k-1} \rho_j \right) \cdot \mathbf{w}_{\mathbf{I}}^{(k)}. \qquad (12)$$

In our experiments, we find that the reasoning chain dependencies are often resolved within one step so we set $K$ small. This keeps the total computational complexity of FLASHTRACE comparable to a single forward pass, while successfully capturing the multi-hop information flow hidden within the reasoning chain. We visualize the shifting attribution distributions across hops in Figure 3.

## 5. Experiments

**Datasets and Tasks.** We evaluate FLASHTRACE on three distinct categories of tasks. **(1) Long-Context Retrieval**: we use the RULER benchmark (Hsieh et al., 2024) to construct Needle-in-a-Haystack (NIAH) and Variable Tracking (VT) tasks at varying context lengths. **(2) Complex Reasoning**: we use MATH (Hendrycks et al., 2021) and MorehopQA (Schnitzler et al., 2024), which require the model to generate intermediate steps before producing a final answer. **(3) Long-Context Multi-hop Reasoning**: we use RULER benchmark to expand HotpotQA samples into long context, which simultaneously tests the model's ability to handle long contexts and perform multi-hop reasoning. Unless otherwise specified, we use Qwen-3 8B Instruct to generate model responses (including $\mathbf{T}$ and output $\mathbf{O}$) and then apply attribution methods to explain the output. For FLASHTRACE, we use $K = 1$ recursive hop in all experiments, as we find this configuration sufficient to resolve the information absorption problem while maintaining efficiency; we analyze the effect of hop count in Appendix H. We compare against established attribution methods; see Appendix D for detailed descriptions. See Appendix E for detailed dataset construction.

**Evaluation Metrics.** We employ two primary metrics to quantify attribution quality. For faithfulness evaluation, following established attribution literature, we use input perturbation metrics: **RISE** (Petsiuk et al., 2018) and **MAS** (Walker et al., 2024), which evaluate how removing or perturbing high-attribution tokens impacts the model's output

*Table 1.* **Attribution quality on RULER benchmarks.** We evaluate Recovery Rate (%, the proportion of ground-truth tokens in top-10% of attribution) and Faithfulness (RISE ↓ and MAS ↓) across context lengths (Needle-in-a-Haystack) and tracking complexity (Variable Tracking).

| Metric | Method | Needle-in-a-Haystack (Long Range Retrieval) | | | | | | Variable Tracking (Synthetic Reasoning) | | | | HotpotQA (Reasoning) |
|---|---|---|---|---|---|---|---|---|---|---|---|---|
| | | mq q2 | mq q4 | mq q8 | mv v2 | mv v4 | mv v8 | h2 c3 | h4 c1 | h6 c1 | h10 c1 | (1024) |
| Recovery Rate (% ↑) | Perturbation | 0.391 | 0.090 | 0.010 | 0.255 | 0.161 | 0.080 | 0.060 | 0.027 | 0.051 | 0.011 | 0.329 |
| | REAGENT | 0.244 | 0.085 | 0.005 | 0.180 | 0.156 | 0.074 | 0.045 | 0.023 | 0.050 | 0.014 | 0.222 |
| | CLP | 0.399 | 0.086 | 0.008 | 0.207 | 0.146 | 0.073 | 0.130 | 0.038 | 0.063 | 0.020 | 0.335 |
| | IFR | 0.471 | 0.328 | 0.012 | **0.575** | 0.452 | 0.179 | 0.136 | 0.253 | 0.202 | 0.155 | 0.268 |
| | AttnLRP | 0.215 | 0.204 | **0.076** | 0.254 | 0.243 | 0.159 | 0.212 | 0.229 | 0.202 | 0.173 | 0.189 |
| | **FLASHTRACE** | **0.483** | **0.413** | 0.075 | 0.556 | **0.516** | **0.204** | **0.698** | **0.755** | **0.659** | **0.514** | **0.384** |
| Faithfulness (RISE ↓) | Perturbation | 0.095 | 0.239 | 0.499 | 0.134 | 0.186 | 0.351 | 0.384 | 0.354 | 0.458 | 0.466 | 0.133 |
| | REAGENT | 0.117 | 0.260 | 0.487 | 0.188 | 0.211 | 0.369 | 0.438 | 0.397 | 0.486 | 0.495 | 0.145 |
| | CLP | 0.098 | 0.253 | 0.510 | 0.156 | 0.217 | 0.393 | 0.374 | 0.328 | 0.423 | 0.451 | 0.101 |
| | IFR | 0.075 | 0.115 | 0.371 | 0.069 | 0.073 | 0.205 | 0.161 | **0.102** | 0.125 | 0.153 | 0.074 |
| | AttnLRP | 0.196 | 0.263 | 0.377 | 0.140 | 0.193 | 0.285 | 0.319 | 0.324 | 0.338 | 0.357 | 0.155 |
| | **FLASHTRACE** | **0.068** | **0.113** | **0.352** | **0.069** | **0.070** | **0.183** | **0.132** | 0.110 | **0.122** | **0.143** | **0.033** |
| Faithfulness (MAS ↓) | Perturbation | 0.144 | 0.327 | 0.709 | 0.187 | 0.244 | 0.458 | 0.551 | 0.517 | 0.684 | 0.701 | 0.220 |
| | REAGENT | 0.197 | 0.357 | 0.694 | 0.276 | 0.291 | 0.494 | 0.668 | 0.603 | 0.745 | 0.741 | 0.235 |
| | CLP | 0.166 | 0.320 | 0.657 | 0.216 | 0.280 | 0.511 | 0.490 | 0.420 | 0.565 | 0.597 | 0.190 |
| | IFR | **0.140** | **0.177** | 0.460 | **0.134** | **0.142** | 0.275 | 0.231 | **0.148** | **0.173** | 0.201 | 0.166 |
| | AttnLRP | 0.326 | 0.451 | 0.602 | 0.229 | 0.325 | 0.475 | 0.521 | 0.548 | 0.572 | 0.592 | 0.249 |
| | **FLASHTRACE** | 0.163 | 0.194 | **0.427** | 0.157 | 0.160 | **0.248** | **0.187** | 0.166 | 0.179 | **0.198** | **0.128** |

*Figure 4.* **Efficiency comparison across methods.** (a-b) Time cost vs. input and generation length. (c-d) Memory consumption vs. input and generation length. (e) Pareto front of speed vs. faithfulness. FLASHTRACE achieves the most efficient scaling in both time and memory, while gradient-based methods (IG, IG-Attn, Perturbation) encounter OOM at longer contexts. Dashed lines indicate OOM.

probability (including $\mathbf{T}$ and output $\mathbf{O}$). For tasks with known ground-truth evidence (i.e., RULER), we propose **Recovery Rate** to calculate the proportion of these ground-truth tokens that appear within the top-10% of the attribution ranking, serving as a direct measure of precision. See Appendix F for detailed metric definitions.

### 5.1. Main Results

**Efficiency in Long-Context Attribution.** We measure the wall-clock time required to perform a single attribution pass on RULER samples of increasing length. We vary both the total context length $|\mathbf{I}|$ and the target span length $|\mathbf{T} + \mathbf{O}|$ to examine scalability along both dimensions. As shown in Figure 4, FLASHTRACE demonstrates significant efficiency gains. At a target span of 5k tokens, FLASHTRACE is over $130\times$ faster than the most efficient baseline (IFR), com-

pleting in under 20 seconds compared to over 38 minutes. [1] Crucially, the span-wise aggregation mechanism allows FLASHTRACE to avoid memory overhead that scales with the target span length. The results validate the suitability of FLASHTRACE for long-context LLM agent interpretation.

**Faithfulness in Long-Context Attribution** The results are summarized in Table 1. In both the direct extraction task (NIAH) and the more complex Variable Tracking (VT) task, FLASHTRACE achieves high Recovery Rates, consistently identifying the ground-truth tokens. While top baselines perform adequately on short, straightforward contexts (i.e. NIAH), their performance degrades as the context length increases. FLASHTRACE maintains superior performance from short contexts up to the maximum evaluated length.

---

[1] Experiments were conducted on a node with 400GB VRAM. See Appendix C for detailed experimental configurations.

The performance margin is especially pronounced on multi-hop reasoning tasks such as HotpotQA. We provide visualizations of typical attribution distributions in Appendix G.

*Table 2.* **Faithfulness on Multi-step Reasoning Tasks.** Comparison on MATH and MorehopQA using causal metrics.

| Method | MATH | | MorehopQA | |
|---|---|---|---|---|
| | RISE ↓ | MAS ↓ | RISE ↓ | MAS ↓ |
| Perturbation | 0.380 | 0.520 | 0.249 | 0.347 |
| REAGENT | 0.388 | 0.536 | 0.265 | 0.368 |
| IFR | 0.354 | 0.490 | 0.146 | 0.228 |
| AttnLRP | 0.368 | 0.524 | 0.286 | 0.458 |
| CLP | 0.362 | 0.491 | 0.249 | 0.348 |
| FLASHTRACE | **0.348** | **0.446** | **0.128** | **0.205** |

**Faithfulness in Multi-step Reasoning.** To test whether FLASHTRACE can handle the "information absorption" problem in reasoning tasks, we evaluate faithfulness on the MATH and MoreHopQA datasets using the RISE and MAS metrics (see Appendix E.4 for dataset construction and sample sizes). In Table 2, FLASHTRACE demonstrates superior faithfulness on both tasks. This suggests that FLASHTRACE effectively captures the flow of importance from the final output, through the intermediate reasoning **T**, back to the input **I**.

**Faithfulness Across Reasoning Lengths** We analyze the impact of length of reasoning tokens on faithfulness by binning the test samples based on the number of generated reasoning tokens. We plot the MAS score against reasoning length in Figure 5. FLASHTRACE maintains stable faithfulness across different reasoning lengths and consistently outperforms baseline methods. This stability confirms that the recursive attribution generalizes effectively to long LLM reasoning and outputs.

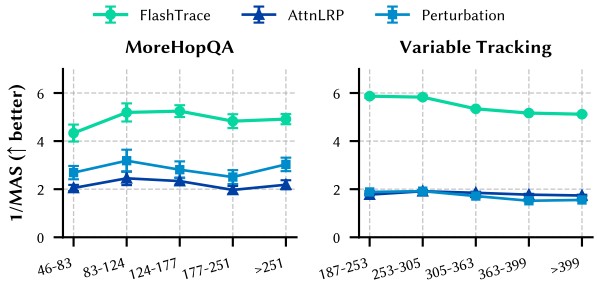

*Figure 5.* Faithfulness (MAS) across reasoning lengths. FLASH-TRACE maintains stable performance and consistently outperforms baseline methods.

## 5.2. Discussion and Ablations

### Compare with Exhaustive Token-Level Rollout

To assess how well FLASHTRACE approximates the complete multi-hop information flow, we compare it against a computationally intensive baseline we term **Exhaustive**

**Token-Level Rollout**. This method represents the theoretical "brute-force" approach to the multi-hop problem: instead of aggregating tokens into spans, it recursively performs attribution for *every* individual reasoning token at each hop (see Appendix D.3 for implementation details). Mathematically, this is equivalent to computing the full product of token-to-token transition matrices without state compression, providing an exact (albeit expensive) trace of influence. Results in Table 3 show that FLASHTRACE matches the performance of this dense baseline on More-hopQA. This indicates that FLASHTRACE successfully captures the critical pathways of information flow in multi-hop reasoning, demonstrating that span-wise aggregation is an effective approximation that avoids the quadratic complexity of exhaustive token-level propagation.

**Generalization to different intermediate content.** We evaluate the generalization of FLASHTRACE to multi-token generation tasks beyond reasoning using the **Aider** dataset, which consists of 133 Python programming exercises. We use Qwen-3 8B to generate code solutions for all exercises. This dataset requires the model to generate code modification snippets based on user queries, where the intermediate content consists of code rather than explicit reasoning steps. We attribute the generated code snippets back to the user request and codebase. Results in Table 4 show that FLASHTRACE consistently outperforms baselines that only attribute the final line. This demonstrates that the multi-token attribution capability of FLASHTRACE generalizes to diverse intermediate content types, making it applicable to a wide range of long-horizon agentic tasks.

*Table 3.* **Approximation Quality vs. Efficiency.** Comparison between FLASHTRACE (Span-wise) and Exhaustive Token-Level Rollout on reasoning tasks. Our method matches the faithfulness of the exact method while reducing runtime from hours to seconds.

| Method | Complexity | Time (s) | RISE ↓ | MAS ↓ |
|---|---|---|---|---|
| Exhaustive Rollout | $\mathcal{O}(M \cdot N)$ | 11.2 | **0.116** | **0.193** |
| FLASHTRACE | $\mathcal{O}(N)$ | 0.72 | 0.128 | 0.205 |
| Delta | *Linear* | ↓*93.6%* | ↑*10.2%* | ↑*6.4%* |

*Table 4.* **Generalization to Code Generation (Aider).** We attribute generated code modification snippets back to user instructions. FLASHTRACE generalizes to structured intermediate outputs better than single-line baselines.

| Method | Target Scope | RISE ↓ | MAS ↓ |
|---|---|---|---|
| IFR (Last Line) | Single Line Output | 0.710 | 0.782 |
| IFR (Token) | Per-Token Avg | 0.707 | 0.773 |
| FLASHTRACE | **Full Code Span** | **0.013** | **0.173** |

**Generalization to different models.** In addition to Qwen-3 8B, we evaluate FLASHTRACE on **LLaMA-3.1-8B-It** (Dubey et al., 2024) to test architectural generalization. Table 5 shows that FLASHTRACE achieves similar performance gains over baselines. This confirms that the method is robust across different model families.

*Table 5.* **Model Generalization to LLaMA-3.1-8B-It.** Faithfulness results on RULER (Avg) and MATH.

| Method | RULER (Avg) | | MATH | |
|---|---|---|---|---|
| | RISE ↓ | MAS ↓ | RISE ↓ | MAS ↓ |
| IFR | 0.206 | 0.298 | 0.435 | 0.603 |
| AttnLRP | 0.398 | 0.683 | 0.473 | 0.686 |
| **FLASHTRACE** | **0.171** | **0.231** | **0.391** | **0.488** |

## 6. Conclusion

We introduced FLASHTRACE, an efficient multi-token attribution method that addresses the efficiency and faithfulness challenges of interpreting reasoning LLMs. By combining span-wise aggregation with recursive attribution, FLASHTRACE achieves efficient span-wise attribution and traces importance through reasoning chains back to source inputs. Experiments across long-context retrieval, mathematical reasoning, and multi-hop QA demonstrate significant speedup while maintaining superior faithfulness, enabling scalable interpretability for modern agentic workflows.

## Acknowledgements

This work was supported in part by the National Natural Science Foundation of China under Grant 62302122, the National Key Research and Development Program of China under Grant 2025YFB3109803, the Heilongjiang Provincial Natural Science Foundation of China under Grant JQ2024F001, and the Hong Kong Research Grants Council under Grants C1043-24GF and RFS2425-1S01.

## Impact Statement

This paper presents work whose goal is to advance the interpretability of large language models. By enabling efficient and faithful attribution of model outputs to their inputs, our method contributes to AI transparency and may help practitioners better understand, debug, and audit model behavior in high-stakes applications. We believe improved interpretability is a positive step toward safer and more trustworthy AI systems. We do not foresee direct negative societal consequences specific to this work.

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

# Appendix

## Table of Contents

# A. Detailed Algorithm of FlashTrace

In this section, we provide a detailed, step-by-step description of the FLASHTRACE algorithm. We formalize the operations performed within the Transformer blocks, the span-wise aggregation mechanism that enables efficiency, and the recursive accumulation process used for multi-hop reasoning tracing.

## A.1. Transformer Architecture Decomposition

We consider a standard Decoder-only Transformer architecture (e.g., Llama, Qwen). A model consists of $L$ layers. The input to the model is a sequence of tokens embedded into vectors $\mathbf{X}^{(0)} = [\mathbf{x}_1, \ldots, \mathbf{x}_{|\mathbf{S}|}]^\top \in \mathbb{R}^{|\mathbf{S}| \times d}$.

For a specific layer $l$, the input state is denoted as $\mathbf{X}^{(l)}$. The layer applies two primary residual blocks: Multi-Head Attention (MHA) and the Multi-Layer Perceptron (MLP).

**Normalization and Linearization.** Modern LLMs typically use RMSNorm or LayerNorm before the attention and MLP blocks. To attribute through these non-linear normalizations, we treat them as element-wise scaling operations. For a vector $\mathbf{x}$, the normalization can be approximated as:

$$\text{Norm}(\mathbf{x}) \approx \mathbf{x} \odot \mathbf{s}(\mathbf{x}), \tag{13}$$

where $\mathbf{s}(\mathbf{x})$ is a scaling vector derived from the running statistics (mean and variance) of $\mathbf{x}$. By fixing $\mathbf{s}$ based on the forward pass activations, we linearize the normalization, allowing us to project contributions directly through the weights.

This element-wise scaling approximation is not introduced by FLASHTRACE; it is an established technique from the ALTI and IFR frameworks (Ferrando et al., 2022; Ferrando & Voita, 2024), which treat the normalization scaling factors as locally constant with respect to the decomposed contributions. FLASHTRACE inherits this treatment without modification, so its linearization fidelity is the same as that of these prior methods rather than a new source of error. We further validate the end-to-end approximation quality empirically: as reported in Table 3, FLASHTRACE closely matches the Exhaustive Token-Level Rollout, which performs the same recursive attribution at full token-level granularity, in both RISE and MAS faithfulness. Because the scaling factors are recomputed from the activations at every layer, the approximation does not compound across depth: each layer is linearized around its own forward-pass statistics, and the recursive hops reuse these same per-layer factors rather than accumulating a global error term.

**Residual Stream Decomposition.** Within layer $l$, the information flow is split into three stages:

1. **Attention Input ($\mathbf{x}^{in}$):** The input to the layer, $\mathbf{x}_i^{(l)}$.

2. **Post-Attention ($\mathbf{x}^{mid}$):** The state after adding the attention output.

$$\mathbf{x}_i^{mid} = \mathbf{x}_i^{in} + \sum_{h=1}^{H} \text{Head}_h(\text{Norm}(\mathbf{x}^{in}))_i. \tag{14}$$

3. **Layer Output ($\mathbf{x}^{out}$):** The state after adding the MLP output.

$$\mathbf{x}_i^{out} = \mathbf{x}_i^{mid} + \text{MLP}(\text{Norm}(\mathbf{x}^{mid}))_i. \tag{15}$$

## A.2. Span-wise Attribution (Single Hop)

The core of FLASHTRACE is the ability to attribute a target span $S = \{s_{start}, \ldots, s_{end}\}$ of $M$ tokens simultaneously in $\mathcal{O}(N)$ time, where $N$ is the total context length.

**1. Aggregated Target Definitions.** Instead of attributing each token $i \in S$ individually, we define aggregated target vectors for the attention and MLP blocks. Let $w_i$ be a scalar weight for token $i$ (for the first hop, $w_i = 1$; for recursive hops, $w_i$ is determined by the previous iteration). The aggregated post-attention target is:

$$\mathbf{X}_S^{mid} = \sum_{i \in S} w_i \cdot \mathbf{x}_i^{mid}. \tag{16}$$

Similarly, the aggregated layer output target is $\mathbf{X}_S^{out} = \sum_{i \in S} w_i \cdot \mathbf{x}_i^{out}$.

**2. Attention Contribution (The Efficiency Key).** Standard attribution requires computing the contribution of source token $j$ to target token $i$, which involves the attention weight $\alpha_{i,j}$. Doing this for all pairs $(i, j)$ where $i \in S$ is computationally expensive. We optimize this by swapping the order of summation. First, we compute the transformed value vector for source token $j$:

$$\mathbf{v}'_j = (\mathbf{x}^{in}_j \odot \mathbf{s}_j)W_V W_O. \tag{17}$$

Here, $W_V$ and $W_O$ are the Value and Output projection matrices. Note that $\mathbf{v}'_j$ depends only on source $j$, not target $i$. The total contribution of source $j$ to the entire span $S$ is:

$$\mathbf{C}_{j \to S} = \sum_{i \in S} w_i \cdot \alpha_{i,j} \cdot \mathbf{v}'_j = \mathbf{v}'_j \cdot \underbrace{\left( \sum_{i \in S} w_i \cdot \alpha_{i,j} \right)}_{\text{Pre-aggregated Attention } A^S_j}. \tag{18}$$

We compute the scalar $A^S_j$ by summing the attention map columns over the rows corresponding to $S$. This reduces the matrix-vector multiplication to a single operation per source token $j$.

**3. Computing Proximity Scores.** We use the L1-based proximity metric $\text{Prox}(\mathbf{c}, \mathbf{t}) = \max(0, -\|\mathbf{t} - \mathbf{c}\|_1 + \|\mathbf{t}\|_1)$. For each layer, we calculate:

- **Token Importance** ($e_{tok,j}$): The proximity of source $j$'s contribution to the target span.

$$e_{tok,j} = \text{Prox}(\mathbf{C}_{j \to S}, \mathbf{X}^{mid}_S). \tag{19}$$

- **Residual Importance** ($e_{res}$): The proximity of the residual stream bypass.

$$\mathbf{R}_S = \sum_{i \in S} w_i \cdot \mathbf{x}^{in}_i \quad \Rightarrow \quad e_{res} = \text{Prox}(\mathbf{R}_S, \mathbf{X}^{mid}_S). \tag{20}$$

- **MLP Importance** ($e_{mlp}$): The proximity of the MLP output to the layer output.

$$\mathbf{M}_S = \sum_{i \in S} w_i \cdot \text{MLP}_i \quad \Rightarrow \quad e_{mlp} = \text{Prox}(\mathbf{M}_S, \mathbf{X}^{out}_S). \tag{21}$$

**4. Normalization and Aggregation.** The raw proximity scores are normalized to form a valid probability distribution. The score for token $j$ at layer $l$ is normalized by the sum of all token contributions plus the residual importance at that layer. We sum these normalized scores across all heads and layers to obtain the global importance distribution $\mathbf{w}^{(k)}$ for the current hop $k$.

**A.3. Recursive Multi-Hop Attribution**

We iterate through $K$ hops to trace influence through the reasoning chain.

**Initialization (Hop 0).** We define the initial target span $S_{out}$ as the model's final output (the answer). We set the weights $w_i = 1$ for all $i \in S_{out}$. We run the span-wise attribution described above to obtain the initial attribution distribution $\mathbf{w}^{(0)}$.

**Recursive Step (Hop $k$).** At step $k$, we focus on explaining the "Thinking Span" (reasoning tokens), denoted as $S_{think}$. The importance of the thinking span in the previous step determines the target for the current step.

1. **Identify Weights:** We extract the importance scores assigned to the thinking tokens in the previous hop:

$$w^{(k)}_t = \mathbf{w}^{(k-1)}[t] \quad \forall t \in S_{think}. \tag{22}$$

2. **Calculate Flow Ratio ($\rho$):** We calculate how much of the total attribution mass is concentrated in the thinking span:

$$\rho_{k-1} = \frac{\sum_{t \in S_{think}} w^{(k)}_t}{\sum_{\text{all } j} \mathbf{w}^{(k-1)}[j]}. \tag{23}$$

3. **Execute Span-Wise Pass:** We run the span-wise attribution with target span $S_{think}$ and token weights $w^{(k)}_t$. This produces a new distribution $\mathbf{w}^{(k)}$ describing which inputs caused the specific reasoning tokens that were important in the previous step.

**Final Accumulation.** The final attribution map is a weighted sum of the distributions from all hops. Since attribution mass flows from Output → Reasoning → Input, we discount subsequent hops by the cumulative ratio of mass that entered the reasoning chain. Let $\mathbf{A}^{(k)}$ be the attribution vector (masked to only include input tokens $\mathbf{I}$) from hop $k$. The final input importance $\mathbf{A}_{final}$ is:

$$\mathbf{A}_{final} = \mathbf{A}^{(0)} + \rho_0 \cdot \mathbf{A}^{(1)} + (\rho_0 \cdot \rho_1) \cdot \mathbf{A}^{(2)} + \dots . \tag{24}$$

This summation ensures that we capture direct influence (Hop 0) as well as indirect influence mediated through the reasoning chain (Hop $k > 0$).

## B. Complexity Analysis of FlashTrace

In this section, we analyze the time and space complexity of FLASHTRACE compared to standard token-level attribution methods. We focus on the attribution phase, assuming the forward pass activations have been cached.

We adopt the following notation throughout this section to ensure clarity:

- $N = |\mathbf{S}|$: the total context length (input + generation)
- $M$: the length of the target span being attributed (e.g., the reasoning chain or final answer)
- $D$: the model's hidden dimension
- $L$: the number of layers

### B.1. Time Complexity

**Naive Token-Level Attribution.** Standard methods (such as Gradient-based Saliency or naive Decomposition) compute attribution scores for each target token individually. To attribute a single target token $i$ to a source token $j$, the method must process the interaction vector in $\mathbb{R}^D$. For a target span of length $M$, the process iterates over all $M$ tokens. For each target, it iterates over all $N$ source tokens. The dominant operation is the vector projection and aggregation (e.g., computing $\|\mathbf{v}_j - \mathbf{y}_i\|_1$).

$$\mathcal{T}_{\text{Naive}} = \mathcal{O}(L \cdot M \cdot N \cdot D). \tag{25}$$

**FLASHTRACE (Single Hop).** FLASHTRACE leverages the linearity of the aggregation to decouple the target span length $M$ from the vector dimension $D$. The process involves two distinct steps per layer: 1. **Scalar Attention Aggregation:** We sum the scalar attention weights $\alpha_{i,j}$ over the target span $S$. This involves iterating over $M$ and $N$, but operates on scalars. Cost: $\mathcal{O}(M \cdot N \cdot 1)$. 2. **Vector Projection:** We compute the transformed value vectors $\mathbf{v}_j$ and scale them by the aggregated weights. Since the weights are pre-aggregated into a single vector $\mathbf{a} \in \mathbb{R}^N$, we only perform vector operations once per source token $j$, independent of $M$. Cost: $\mathcal{O}(N \cdot D)$.

Combining these, the complexity for one hop is:

$$\mathcal{T}_{\text{FlashTrace}} = \mathcal{O}\left(L \cdot (M \cdot N + N \cdot D)\right) = \mathcal{O}\left(L \cdot N \cdot (M + D)\right). \tag{26}$$

Since $D$ is typically large (e.g., 4096), the reduction from $M \cdot N \cdot D$ to $N \cdot D$ represents a significant speedup. Note that while the $\mathcal{O}(M \cdot N)$ scalar aggregation term remains, it involves only simple floating-point additions and is negligible compared to the $D$-dimensional vector operations in practice.

**Multi-Hop Overhead.** For $K$ recursive hops, the cost scales linearly:

$$\mathcal{T}_{\text{Total}} = K \cdot \mathcal{T}_{\text{FlashTrace}} \approx \mathcal{O}(K \cdot L \cdot N \cdot D). \tag{27}$$

Since $K$ is a small constant (typically $2 - 4$), the total time remains linear with respect to context length $N$ and independent of the multiplicative interaction between $M$ and $D$.

### B.2. Space Complexity

**Naive Approach.** To vectorize operations, naive implementations often materialize a tensor of shape $[M, N, D]$ to store the pairwise contributions before reduction, or compute gradients of shape $[M, N]$.

$$\mathcal{M}_{\text{Naive}} \approx \mathcal{O}(M \cdot N \cdot D) \quad \text{or} \quad \mathcal{O}(M \cdot N) \text{ (gradients)}. \tag{28}$$

**Algorithm 1** FLASHTRACE: Efficient Multi-Hop Attribution

---

**input** Context consisting of Input $\mathbf{I}$, Reasoning $\mathbf{T}$, and Output $\mathbf{O}$. Model $\mathcal{M}$ with $L$ layers. Number of hops $K$.
**output** Token-level attribution scores $\mathbf{A}_{final}$ for Input $\mathbf{I}$.
1: **Forward Pass:** Run $\mathcal{M}(\mathbf{I} \circ \mathbf{T} \circ \mathbf{O})$. Cache hidden states $\mathbf{X}^{in}, \mathbf{X}^{mid}, \mathbf{X}^{out}$ and Attention maps $\mathbf{A}$ for all layers.
2: Initialize attribution list $\mathcal{W}_{list} \leftarrow []$, flow ratios $\rho_{list} \leftarrow []$.
3: **function** SPANATTRIBUTE(Target Span $S$, Token Weights $w$)
4:     Initialize global importance scores $\mathbf{E} \leftarrow \mathbf{0}^{|\mathbf{I}|+|\mathbf{T}|+|\mathbf{O}|}$.
5:     **for** layer $l = 1$ **to** $L$ **do**
6:         {1. Aggregate Target Representations (Span-wise)}
7:         $\mathbf{Y}_S^{mid} \leftarrow \sum_{i \in S} w_i \mathbf{X}_i^{mid}$
8:         $\mathbf{Y}_S^{out} \leftarrow \sum_{i \in S} w_i \mathbf{X}_i^{out}$
9:         {2. Efficient Attention Attribution ($\mathcal{O}(N)$ complexity)}
10:        Calculate pre-aggregated attention weights:
11:        $\boldsymbol{\alpha}^S \leftarrow \sum_{i \in S} w_i \mathbf{A}_{i,:}^{(l)}$    {Summing rows of attention map}
12:        **for** source token $j \in$ Context **do**
13:            Compute independent value vector: $\mathbf{v}_j' \leftarrow (\mathbf{X}_j^{in} \odot \mathbf{s}_j) W_V W_O$
14:            Compute contribution to span: $\mathbf{C}_{j \rightarrow S} \leftarrow \boldsymbol{\alpha}_j^S \cdot \mathbf{v}_j'$
15:            Compute proximity: $e_j \leftarrow \text{Proximity}(\mathbf{C}_{j \rightarrow S}, \mathbf{Y}_S^{mid})$
16:            $\mathbf{E}[j] \leftarrow \mathbf{E}[j] + e_j$
17:        **end for**
18:        {3. Residual and MLP Attribution}
19:        Calculate $e_{res} \leftarrow \text{Proximity}(\sum_{i \in S} w_i \mathbf{X}_i^{in}, \mathbf{Y}_S^{mid})$
20:        Calculate $e_{mlp} \leftarrow \text{Proximity}(\sum_{i \in S} w_i \text{MLP}(\mathbf{X}_i^{mid}), \mathbf{Y}_S^{out})$
21:        Normalize layer scores $\mathbf{E}$ using $e_{res}$ and $e_{mlp}$.
22:     **end for**
23:     **return** Normalized distribution $\mathbf{w}$
24: **end function**
25: {Hop 0: Attribute Output $\mathbf{O}$ to Context}
26: $\mathbf{w}^{(0)} \leftarrow$ SPANATTRIBUTE$(\mathbf{O}, \mathbf{1}_{|\mathbf{O}|})$
27: $\mathcal{W}_{list}.\text{append}(\mathbf{w}^{(0)})$
28: {Recursive Hops: Trace through Reasoning $\mathbf{T}$}
29: **for** hop $k = 1$ **to** $K$ **do**
30:     Extract weights for reasoning span: $w_{next} \leftarrow \{\mathbf{w}_t^{(k-1)} \mid t \in \mathbf{T}\}$
31:     Calculate flow ratio: $\rho \leftarrow (\sum_{t \in \mathbf{T}} \mathbf{w}_t^{(k-1)})/(\sum_j \mathbf{w}_j^{(k-1)})$
32:     $\rho_{list}.\text{append}(\rho)$
33:     {Attribute weighted reasoning span back to context}
34:     $\mathbf{w}^{(k)} \leftarrow$ SPANATTRIBUTE$(\mathbf{T}, w_{next})$
35:     $\mathcal{W}_{list}.\text{append}(\mathbf{w}^{(k)})$
36: **end for**
37: {Final Aggregation: Sum weighted contributions}
38: $\mathbf{A}_{final} \leftarrow \mathcal{W}_{list}[0][\mathbf{I}]$
39: accum_rho $\leftarrow 1.0$
40: **for** hop $k = 1$ **to** $K$ **do**
41:     accum_rho $\leftarrow$ accum_rho $\times \rho_{list}[k-1]$
42:     $\mathbf{A}_{final} \leftarrow \mathbf{A}_{final} +$ accum_rho $\cdot \mathcal{W}_{list}[k][\mathbf{I}]$
43: **end for**
44: **return** $\mathbf{A}_{final}$

---

This leads to Out-Of-Memory (OOM) errors for long contexts, as observed in our experiments (Figure 4).

**FLASHTRACE.** FLASHTRACE processes the attention matrix in blocks (chunking). We never materialize the full pairwise interaction tensor.

1. We store the forward cache (Key/Value states), which is inherent to inference: $\mathcal{O}(N \cdot D)$.

2. For attribution, we store the pre-aggregated attention weights: $\mathcal{O}(N)$.

3. We store the aggregated target vector: $\mathcal{O}(D)$.

4. During chunked computation, we process small blocks of size $B$. Memory usage is $\mathcal{O}(B \cdot D)$.

Thus, the peak *working memory* footprint during the attribution computation (excluding the fixed model weights, KV cache, and attention maps) is:

$$\mathcal{M}_{\text{FlashTrace}}^{\text{work}} = \mathcal{O}(N + D). \tag{29}$$

This working memory is independent of the target span length $M$, allowing efficient processing of long generated sequences without additional memory overhead per target token.

**Attention Cache Requirement.** Note that the algorithm requires access to the attention maps across all layers (Algorithm 1, Line 1), which necessitates $\mathcal{O}(L \cdot N^2)$ space if stored explicitly. In practice, this can be mitigated by recomputing attention on-the-fly or using memory-efficient attention implementations that trade compute for memory.

### B.3. Summary of Improvements

Table 6 summarizes the theoretical comparison. FLASHTRACE shifts the heavy vector arithmetic ($\cdot D$) out of the loop over the target span $M$. Although a residual $\mathcal{O}(M \cdot N)$ term remains for scalar attention weight aggregation, this lightweight operation is negligible in practice, resulting in linear scaling for long-context attribution.

*Table 6.* **Complexity Comparison.** $N$: Context Length, $M$: Target Span Length, $D$: Hidden Dimension, $K$: Number of Hops, $L$: Number of Layers. Working memory refers to memory used during attribution computation, excluding attention maps (which require $\mathcal{O}(L \cdot N^2)$ if cached).

| Method | Time Complexity | Working Memory |
|---|---|---|
| Naive Token-Level | $\mathcal{O}(M \cdot N \cdot D)$ | $\mathcal{O}(M \cdot N)$ |
| **FLASHTRACE (Ours)** | $\mathcal{O}(\mathbf{K} \cdot \mathbf{N} \cdot (\mathbf{M} + \mathbf{D}))$ | $\mathcal{O}(\mathbf{N} + \mathbf{D})$ |

## C. Efficiency Experiment Details

In this section, we provide detailed experimental configurations for the efficiency comparison presented in Figure 4.

**Hardware Configuration.** All efficiency experiments were conducted on a compute node equipped with 5× NVIDIA A800 GPUs (80GB VRAM each), providing a total of 400GB GPU memory.

**Sequence Length Configurations.** To systematically evaluate scalability, we vary both the input context length $|\mathbf{I}|$ and the target generation length $|\mathbf{T} + \mathbf{O}|$ across the following values: 10, 100, 500, 1000, 2000, and 5000 tokens. When varying one dimension, the other is held constant at 100 tokens. Specifically, in Figures 4(a) and (c), we fix the generation length at 100 tokens and vary the input length. In Figures 4(b) and (d), we fix the input length at 100 tokens and vary the generation length.

**Memory Measurement.** Memory consumption reported in Figures 4(c) and (d) corresponds to *peak GPU memory usage* during the attribution process, measured using PyTorch's `torch.cuda.max_memory_allocated()`.

**Notes on IG and IG-Attn in the Pareto Plot.** In Figure 4(e), Integrated Gradients (IG) and Attention-aware Integrated Gradients (IG-Attn) are positioned at the leftmost side of the plot. This placement reflects that we were unable to evaluate their faithfulness metrics in our experimental setting. Both methods encountered out-of-memory (OOM) errors on the majority of samples when processing the sequence lengths required for our RULER benchmark evaluation. As a result, we could not compute reliable RISE and MAS scores for these methods. For transparency, we include them in the plot to show their relative computational cost, but their horizontal position should not be interpreted as a faithfulness measurement. For all other methods, we successfully computed faithfulness metrics with average MAS on RULER datasets.

# D. Baseline Methods

We compare FLASHTRACE against a diverse set of established attribution methods. Baseline implementations for IG, IG-Attn, Perturbation, REAGENT, CLP are adapted from implementation by Walker & Ewetz (2025). IFR and AttnLRP are implemented based on open-source implementation provided by the original authors.

**Integrated Gradients (IG).** Adapted for language models by Vafa et al. (2021) from Sundararajan et al. (2017), IG computes attribution by integrating gradients along a linear path from a baseline zero-embedding to the actual input. This method satisfies axiomatic properties such as sensitivity and completeness.

**Attention-aware Integrated Gradients (IG-Attn).** Chen et al. (2022) propose a framework that combines attention mechanisms with gradient signals. By applying Integrated Gradients specifically to attention maps rather than input embeddings, it decomposes attribution into attention perception and gradient-based reasoning feedback.

**Perturbation-based Attribution.** We implement the standard Leave-One-Out (LOO) strategy (Liu et al., 2024), which serves as a causal baseline. This method estimates token importance by masking individual input tokens and measuring the resulting drop in the log-probability of the target output.

**REAGENT.** Zhao & Shan (2024) introduce a model-agnostic perturbation method for generative tasks. Instead of masking tokens, REAGENT replaces selected tokens with plausible alternatives generated by an auxiliary model (e.g., RoBERTa) and measures the output distribution shift using Hellinger distance.

**Context Length Probing (CLP).** Cífka & Liutkus (2023) propose a black-box method that assigns importance by systematically varying the context length. It calculates the differential contribution of a token by observing how the prediction confidence changes when that token is added to the context window.

**Information Flow Routes (IFR).** Based on the ALTI framework (Ferrando et al., 2022), IFR (Ferrando & Voita, 2024) decomposes the transformer computation into a directed graph. It traces information flow by calculating the proximity of token vectors through attention heads and MLP blocks layer by layer to quantify contribution.

**AttnLRP.** Achtibat et al. (2024) adapt Layer-wise Relevance Propagation to Transformer architectures. This method defines specific propagation rules for non-linear components like Softmax and LayerNorm, allowing relevance scores to be backpropagated from the output to the input while maintaining conservation properties.

## D.1. Extending Baseline Methods to Multi-Token Attribution

The baseline methods described above are fundamentally designed for single-token attribution. They calculate the influence of the context on a specific output token $o_t$ at a single timestep. However, the tasks in our evaluation involve multi-token outputs, such as long chains of reasoning or complete code snippets. To compare these baselines fairly with FLASHTRACE, we extend them using an iterative aggregation strategy.

Specifically, for a generated output sequence $\mathbf{O}$ (excluding the intermediate reasoning tokens $\mathbf{T}$), we treat every token $o_t \in \mathbf{O}$ as an independent attribution target. We iterate through the output sequence, computing an attribution vector $\mathbf{a}^{(t)}$ for each token $o_t$ separately. We then average these individual vectors to obtain a global importance distribution $\mathbf{A}_{global}$:

$$\mathbf{A}_{global} = \frac{1}{|\mathbf{O}|} \sum_{t=1}^{|\mathbf{O}|} \mathbf{a}^{(t)}. \tag{30}$$

As discussed in Section 3.2, standard attribution methods tend to assign a significant portion of the importance mass to the immediate history—in this case, the intermediate reasoning tokens $\mathbf{T}$—rather than the original input $\mathbf{I}$. This creates a distribution mismatch when evaluating input faithfulness. To address this, we apply a post-processing step for all baseline methods during evaluation. We discard the attribution scores assigned to the reasoning tokens $\mathbf{T}$ and the output tokens $\mathbf{O}$, retaining only the scores assigned to the input $\mathbf{I}$. We then re-normalize this truncated vector to form a valid probability distribution over the input:

$$\mathbf{A}_{input}[i] = \frac{\mathbf{A}_{global}[i]}{\sum_{j \in \mathbf{I}} \mathbf{A}_{global}[j]} \quad \forall i \in \mathbf{I}. \tag{31}$$

This ensures that the evaluation metrics (RISE, MAS, and Recovery Rate) measure the relative ranking of the source information strictly within the input prompt.

### D.2. Optimizing Baseline Methods for Long Contexts Efficiency

Perturbation-based methods (specifically Perturbation, REAGENT, and CLP) present a unique challenge in long-context settings. By default, these methods operate at the token level, ablating or altering one token at a time to measure the effect on the output. For a context of length $|\mathbf{S}|$, this approach requires $\mathcal{O}(|\mathbf{S}|)$ forward passes. In our RULER experiments, where contexts can reach 32k tokens, running tens of thousands of forward passes for a single attribution sample is computationally prohibitive.

To make these methods tractable for long-context benchmarks, we adopt a coarse-grained perturbation strategy. Following the implementation in Walker & Ewetz (2025), we shift the granularity from tokens to sentences (or text chunks). Instead of masking individual tokens, we segment the input text into sentences. We then systematically mask each sentence and measure the change in the model's output probability. The importance score derived for a sentence is then assigned uniformly to all its constituent tokens. This optimization reduces the number of forward passes from the number of tokens to the number of sentences, typically reducing runtime by a factor of 20-50$\times$ while preserving sufficient granularity to identify key information in retrieval tasks.

### D.3. Exhaustive Token-Level Rollout Implementation

To provide an upper-bound comparison for FLASHTRACE's span-wise approximation, we implement an **Exhaustive Token-Level Rollout** baseline. This method serves as a computationally expensive but theoretically exact approach to multi-hop attribution.

**Method Description.** The exhaustive rollout performs recursive attribution by first attributing the output to each individual reasoning token, then attributing those important reasoning tokens back to the input. This process iterates to trace the complete multi-hop information flow at token-level granularity. Concretely, at each recursive step, we run attribution for *every individual reasoning token* rather than aggregating them into spans. This approach is conceptually similar to CAGE (Walker & Ewetz, 2025), a concurrent work that also explores multi-hop attribution for explaining LLM reasoning chains; however, CAGE operates at sentence-level granularity for computational tractability, whereas our exhaustive rollout maintains token-level precision.

**Attribution Method.** We apply the exhaustive rollout to IFR (Information Flow Routes) (Ferrando & Voita, 2024), which achieves the strongest baseline performance in our evaluation. IFR provides a principled decomposition of transformer computations that is well-suited for recursive attribution.

**Complexity.** This token-level exhaustive approach is significantly more expensive than FLASHTRACE, requiring $\mathcal{O}(M)$ attribution passes per hop, where $M = |\mathbf{T}|$ is the number of reasoning tokens. However, it provides the most fine-grained recursive attribution possible and serves as a theoretical upper bound for approximation quality.

**Evaluation Setting.** For the comparison in Table 3, we evaluate on the MoreHopQA dataset using the same 100 samples with model-generated reasoning as other experiments (see Appendix E.4).

## E. Dataset Construction

In this section, we provide detailed construction methodology for the four datasets used to evaluate FLASHTRACE. These datasets cover diverse reasoning scenarios: long-context retrieval, mathematical problem-solving, multi-hop question answering, and code generation. For all datasets, we format the data into the Input $\mathbf{I}$, Reasoning/Intermediate $\mathbf{T}$, and Output $\mathbf{O}$ structure described in Section 3.

### E.1. Long-Context Retrieval: RULER

To assess attribution performance in long-context scenarios, we utilize the RULER benchmark (Hsieh et al., 2024), which provides synthetic tasks with controllable complexity and context length. We focus on three specific tasks:

**Needle-in-a-Haystack (NIAH).**    This task tests the model's ability to retrieve specific information hidden within a large context. We construct samples by inserting "needles" (key-value pairs in the format "The special magic number for $X$ is $Y$") into a "haystack" of distractor text (Paul Graham essays). We vary the context length from 4k to 32k tokens. For attribution evaluation, the **Ground Truth** corresponds to the tokens comprising the inserted needle sentence. A faithful attribution method should assign high importance scores to these specific tokens within the long distractor context.

**Variable Tracking (VT).**    This task simulates multi-hop reasoning by requiring the model to trace value assignments through a chain of variables (e.g., $X_1 = V, X_2 = X_1, \dots$). We construct chains with varying numbers of hops (binding depth) and parallel distractor chains. The input consists of the variable assignment statements dispersed throughout noise text. The **Ground Truth** for attribution includes all variable assignment statements involved in the target chain. This task is particularly challenging for attribution as the model must identify the complete dependency path across the context.

**Long-Context HotpotQA.**    To evaluate realistic multi-hop reasoning, RULER adapts samples from the HotpotQA dataset. Each sample includes a question and a set of paragraphs. The "golden" paragraphs containing the answer are treated as needles, while distractor paragraphs are sampled to fill the context to the target length. The **Ground Truth** consists of the tokens belonging to the golden paragraphs that support the answer.

**Task Configuration Abbreviations.**    Table 1 uses abbreviated codes to denote different task configurations from the RULER benchmark. We clarify these abbreviations below:

- **Multi-Query NIAH (mq):** Tests retrieval of multiple distinct key-value pairs. The suffix indicates the number of queries: `mq q2`, `mq q4`, and `mq q8` require retrieving 2, 4, and 8 different key-value pairs, respectively.

- **Multi-Value NIAH (mv):** Tests retrieval of multiple values associated with a single key. The suffix indicates the number of values: `mv v2`, `mv v4`, and `mv v8` require retrieving 2, 4, and 8 values per key, respectively.

- **Variable Tracking (VT):** Tests multi-hop tracing through variable binding chains. The notation $hX$ $cY$ indicates $X$ hops (binding depth) and $Y$ parallel chains. For example, `h6 c1` denotes a single chain with 6 variable bindings, while `h2 c3` denotes 3 parallel chains with 2 bindings each.

**Context Length Configuration.**    For all RULER benchmark experiments reported in Table 1, we use a fixed context length of 1024 tokens across all NIAH and VT task variants. This standardized configuration ensures fair comparison across different attribution methods while maintaining computational efficiency.

### E.2. Complex Reasoning: MoreHopQA and MATH

These datasets are selected to evaluate the model's ability to perform multi-step reasoning and generation, where the intermediate reasoning **T** plays a crucial role.

**MoreHopQA.**    Schnitzler et al. (2024) introduced this dataset to prevent models from exploiting shortcuts in standard multi-hop QA. It extends 2-hop questions into 3+ hop questions using arithmetic, symbolic, or commonsense reasoning templates. For example, a question about a birth date might be extended to ask for the date "one week after" that birth date. The dataset contains 1,118 human-verified samples. Each sample is annotated with explicit provenance (supporting paragraphs) for every reasoning hop.

**MATH.**    The MATH dataset (Hendrycks et al., 2021) consists of 12,500 competition-level mathematics problems (AMC 10, AMC 12, AIME). Unlike simple arithmetic tasks, these problems require constructing a proof or a sequence of logical steps. We use the problem statement as the Input **I**. The model generates a step-by-step solution in LaTeX format, which serves as the Reasoning **T**, culminating in a boxed final answer **O**. Since there are no explicit token-level labels for "evidence" in math problems, we evaluate attribution faithfulness using perturbation metrics (RISE/MAS) rather than recovery rate.

### E.3. Code Generation: Aider

To test generalization to structured output generation, we use the Aider Code Editing Benchmark (Gauthier, 2023). This benchmark consists of 133 Python exercises sourced from the Exercism platform. Each sample includes a natural language instruction describing the task and a "stub" code file with function signatures. This task differs from pure reasoning as the

"intermediate" tokens are structured code rather than natural language thoughts. We attribute the generated code body back to the instructions and function definitions in the input.

### E.4. Generating Intermediate Reasoning Tokens T

While the datasets described above provide the problem input $\mathbf{I}$ and the ground-truth answer $\mathbf{O}$, they do not natively contain the model-generated intermediate reasoning traces $\mathbf{T}$ required for our analysis. To evaluate multi-token attribution in a realistic agentic setting, we must first generate these reasoning steps using the target model.

Specifically, we employ Qwen-3 8B Instruct to perform inference on all dataset samples. We capture the complete generation context, consisting of the original user prompt $\mathbf{I}$, the generated reasoning $\mathbf{T}$, and the final answer $\mathbf{O}$. To ensure the validity of our interpretability analysis, we filter these generations to retain only those where the model produced a correct final answer. Attributing the reasoning process of an incorrect prediction may introduce noise, as the "reasoning" itself is likely flawed or hallucinatory.

From this filtered pool, we randomly select a subset of 100 correctly answered samples for each dataset to serve as our evaluation set. An exception is the long-context HotpotQA task, which is significantly more difficult; for this dataset, we utilize the 66 samples where the model successfully retrieved the correct answer.

## F. Evaluation Details

We employ two complementary approaches to evaluate attribution quality: ground-truth recovery for synthetic tasks, and perturbation-based faithfulness metrics for complex reasoning tasks where exact ground-truth labels are unavailable.

### F.1. Recovery Rate

For tasks in the RULER benchmark (Needle-in-a-Haystack and Variable Tracking), the dataset construction process explicitly inserts "needle" sentences or variable definitions into a "haystack" of noise. These inserted segments constitute the ground-truth evidence set $S_{gt} \subset \mathbf{I}$.

To measure how well an attribution method identifies these known causal factors, we define the **Recovery Rate**. Let $K = \lfloor 0.1 \times |\mathbf{I}| \rfloor$. We identify the set of top-$K$ tokens $S_{top}$ with the highest attribution scores. The Recovery Rate is defined as the recall of the ground-truth tokens within this top-10% bracket:

$$\text{Recovery Rate} = \frac{|S_{top} \cap S_{gt}|}{|S_{gt}|}. \tag{32}$$

A higher recovery rate indicates that the attribution method successfully ranks the true causal inputs above the distractor context.

### F.2. Perturbation Metrics: RISE and MAS

For datasets like MATH and MoreHopQA, there are no explicit token-level labels indicating which parts of the input are "correct." In these cases, we rely on established faithfulness metrics: **RISE** (Petsiuk et al., 2018) and **MAS** (Walker et al., 2024).

Both metrics are perturbation-based: they measure the quality of an attribution map by iteratively masking input tokens based on their assigned importance and observing the degradation in the model's confidence. In our setting, the target confidence is the joint probability of the generated sequence (including both reasoning $\mathbf{T}$ and output $\mathbf{O}$) conditioned on the perturbed input.

**Deletion Test.** We specifically employ the Deletion test, where high-importance tokens are removed first. The intuition is that if an attribution method is faithful, removing the most important tokens should cause the most rapid drop in model probability. For Deletion metrics, **lower is better**.

**Metric Definitions.** Let $f(\mathbf{I})_c$ represent the model's probability for the target sequence given input $\mathbf{I}$. Let $\pi$ be the permutation of input indices that sorts the attribution scores in descending order.

- **RISE Deletion:** This metric calculates the area under the probability curve (AUC) as tokens are masked.

$$\text{RISE}_{\text{del}} = \frac{1}{N} \sum_{k=1}^{N} f(\mathbf{I}_{\text{masked}}^{(k)})_c, \tag{33}$$

  where $\mathbf{I}_{\text{masked}}^{(k)}$ is the input with the top-$k$ most important tokens replaced by a baseline token (e.g., a padding token).

- **MAS Deletion:** The Magnitude Aligned Scoring (MAS) metric improves upon RISE by penalizing attributions where the magnitude of the score does not align with the magnitude of the model's response. It adds an alignment penalty term:

$$\text{MAS}_{\text{del}} = \text{RISE}_{\text{del}} + \frac{1}{N} \sum_{k=1}^{N} \left| f(\mathbf{I}_{\text{masked}}^{(k)})_c - \frac{\sum_{i=1}^{k} |A_{\pi(i)}|}{\sum_j |A_j|} \right|. \tag{34}$$

  This ensures that attribution methods are sensitive to the absolute scale of importance, not just the ranking.

**Efficient Evaluation.** Standard implementations of RISE and MAS evaluate the model after masking each individual token (i.e., step size of 1). For long-context inputs with thousands of tokens, this would require thousands of forward passes per sample, which is computationally infeasible.

To adapt these metrics for the long-context setting, we modify the perturbation schedule to use a proportional step size. Instead of masking one token at a time, we mask the top 5% of the remaining tokens at each step. This discretizes the evaluation into exactly 20 intervals: $k \in \{0.05|\mathbf{I}|, 0.10|\mathbf{I}|, \ldots, |\mathbf{I}|\}$. This modification allows us to compute robust faithfulness metrics for any context length using a fixed budget of 20 forward passes per sample.

## G. Case Study Visualizations

In addition to the MoreHopQA examples presented in the main text, we provide visualizations on the MATH dataset in Figure 6. Comparing the attribution distributions between Hop 1 and Hop 2, we observe that key information in the input region is strengthened, while the attribution mass on the reasoning region is attenuated. This demonstrates how recursive attribution successfully traces influence back through the reasoning chain to the source inputs.

## H. Ablation: Effect of Recursive Hops

We examine the impact of recursion depth $K$ on attribution quality. Table 7 shows faithfulness metrics across different hop configurations on the MorehopQA dataset.

Direct attribution ($K = 0$) performs poorly as reasoning tokens absorb the majority of importance. Introducing recursion ($K = 1$) significantly improves faithfulness by redistributing influence upstream. We find that performance remains stable across hop counts, suggesting that critical causal dependencies are effectively resolved within the first recursive step. Extending to higher hop counts yields diminishing returns, likely due to noise accumulation in the attribution signal.

*Table 7.* **Effect of Recursive Hops.** Faithfulness (RISE ↓, MAS ↓) on MorehopQA with varying recursion depths.

| Configuration | RISE ↓ | MAS ↓ |
|---|---|---|
| No Recursion (0 Hops) | 0.127 | 0.209 |
| 1 Hop | **0.128** | **0.205** |
| 2 Hops | 0.130 | 0.206 |
| 3 Hops | 0.133 | 0.209 |

**Attribution Mass Distribution Across Hops.** To further illustrate the effect of recursive attribution in propagating influence from $\mathbf{T}$ back to $\mathbf{I}$, we visualize the proportion of attribution mass assigned to different token segments at each hop.

As shown in Figure 7, the attribution mass concentrated on reasoning tokens decreases from hop 0 to hop 1, while attribution shifts toward input tokens. Within the reasoning span itself, attribution also shifts towards earlier reasoning tokens. This confirms that recursive attribution successfully redistributes the information absorbed by reasoning tokens back to the source inputs.

## I. Pilot Study Details

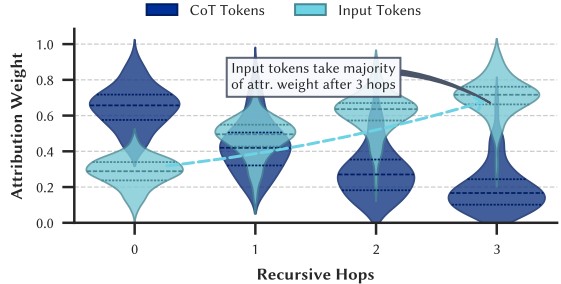

*Figure 6.* Token-level attribution heatmaps on MATH dataset samples. **Hop 1** attributes the output directly to the context. **Hop 2** recursively attributes through the reasoning chain. Δ **(Hop2 − Hop1)** shows the change: green indicates increased attribution (input tokens), red indicates decreased attribution (reasoning tokens).

We conduct our pilot study using the multi-hop retrieval task from RULER (Hsieh et al., 2024)'s VT subset. Specifically, we use one example from the validation set and prompt Qwen-3 8B Instruct under two configurations: (1) *direct answering*, where the model is instructed to output the answer immediately without explicit reasoning, and (2) *chain-of-thought* (CoT), where the model is encouraged to generate step-by-step reasoning before providing the final answer. The "Baseline" method shown in Figure 1(b) refers to the perturbation-based attribution approach described in Section D.

*Figure 7.* Attribution mass shift during recursive hops. From hop 0 to hop 1, the attribution mass on reasoning tokens decreases while attribution shifts toward input tokens. Aggregated across 100 samples from the MoreHopQA dataset.

## J. Additional Related Work

This section expands on related interpretability approaches that we could only briefly mention in Section 2.

**Feature-Circuit Interpretability.** A complementary family of interpretability methods analyzes internal feature circuits rather than input-token importance. Transcoders decompose the MLP sublayers of a model into dictionaries of interpretable features (Dunefsky et al., 2024), and circuit-tracer identifies the minimal set of features and connections responsible for a specific behavior (Hanna et al., 2025). These methods answer the question of *which internal components* implement a behavior, operating at the feature-circuit level. FLASHTRACE answers a different question, namely *which input tokens* inform a target span, operating at the token level over the information that flows through attention and residual connections. The two paradigms are complementary rather than competing. A natural workflow is to first apply FLASHTRACE to localize the input regions that matter, which is cheap even for long contexts, and then apply feature-circuit analysis within those regions for a mechanistic account of how the model uses them.

## K. Additional Faithfulness Experiments

The faithfulness evaluation in the main text relies on two design choices that warrant further analysis: it filters generations to those the model answers correctly, and it fixes the RULER context length at 1024 tokens. This section reports additional experiments that relax both choices.

### K.1. Faithfulness without Correctness Filtering

The correctness filter described in Appendix E.4 is required only by the Recovery Rate metric, which measures whether the attribution recovers the ground-truth evidence and is therefore only meaningful when the model has actually used that evidence to answer correctly. The perturbation metrics RISE and MAS carry no such requirement: they measure how the model's output probability responds to masking high-attribution tokens, regardless of whether the answer is right. FLASHTRACE itself is also independent of correctness, since it traces information flow through the realized generation whether or not that generation is correct. To verify that our conclusions are not an artifact of filtering, we re-evaluate on HotpotQA with no correctness filter, reporting only RISE and MAS.

### K.2. Faithfulness at Longer Contexts

We additionally extend the HotpotQA evaluation from the 1024-token setting used in Table 1 to 2048 and 4096 tokens, which directly tests whether the faithfulness advantage holds as the context grows. Table 8 reports both experiments together. FLASHTRACE retains a clear advantage over IFR and AttnLRP in the unfiltered setting, confirming that the gains reported in the main text are not an effect of correctness filtering. Moreover, its faithfulness improves rather than degrades at longer contexts, with RISE dropping from 0.036 at 2048 tokens to 0.025 at 4096 tokens. This indicates that span-wise recursive attribution remains reliable in exactly the long-context regime it is designed for.

*Table 8.* **Unfiltered faithfulness at longer contexts.** RISE and MAS on HotpotQA with no correctness filtering, at 2048 and 4096 token contexts. FLASHTRACE keeps its advantage and improves at longer contexts.

| Context | Method | RISE ↓ | MAS ↓ | Time (s) |
|---|---|---|---|---|
| | AttnLRP | 0.383 | 0.544 | 1.19 |
| 2048 | IFR | 0.059 | 0.139 | 5.10 |
| | **FLASHTRACE** | **0.036** | **0.118** | 3.58 |
| | AttnLRP | 0.232 | 0.291 | 2.63 |
| 4096 | IFR | 0.040 | 0.115 | 12.71 |
| | **FLASHTRACE** | **0.025** | **0.106** | 9.12 |

# L. Memory and Runtime Trade-offs

Algorithm 1 caches the attention maps of all layers during the forward pass, which costs $\mathcal{O}(L \cdot N^2)$ space as noted in Appendix B.2. For memory-constrained deployments this cache can be avoided entirely. Because FLASHTRACE consumes the attention maps in the same layer order as the forward pass, each layer's attention can be recomputed on the fly at the moment it is needed and then discarded, trading additional compute for a smaller memory footprint.

We benchmark both modes on Qwen-3 4B (bf16) with a 2048-token context, averaging over five runs. Table 9 reports the result. The on-the-fly mode reduces peak GPU memory by 62%, from 35.43 GiB to 13.43 GiB, at the cost of a 42% increase in latency. Users can therefore select the mode that matches their hardware: caching is preferable when memory is ample, while on-the-fly recomputation makes FLASHTRACE viable on consumer-grade GPUs.

*Table 9.* **Cached versus on-the-fly attention.** Latency and peak GPU memory of FLASHTRACE on Qwen-3 4B (bf16, 2048-token context, averaged over five runs).

| Mode | Time (s) | Peak GPU Memory (GiB) |
|---|---|---|
| Cached attention maps | $1.98 \pm 0.27$ | 35.43 |
| On-the-fly recomputation | $2.81 \pm 0.06$ | 13.43 |

# M. Alternative Span Aggregation Strategies

FLASHTRACE aggregates a target span by a (weighted) sum of its token representations, as defined in Eq. (5). This section examines alternative aggregation choices and the granularity at which spans are constructed.

## M.1. Aggregation Operators

**Mean pooling.** Replacing the sum with a mean over individual token attributions corresponds to the IFR baseline, which is effectively the single-token fallback of FLASHTRACE applied to each target token and then averaged. Crucially, mean pooling performs the aggregation *after* the non-linear L1 proximity operation, which breaks the linearity that FLASHTRACE exploits to pre-aggregate attention weights (Section 4.2). It therefore loses the single-pass efficiency of span-wise aggregation and is substantially slower, while as Table 1 shows it also recovers fewer ground-truth tokens.

**Last-token targeting.** Another alternative is to attribute only the final token of the generation rather than the full span. Table 10 compares this choice against FLASHTRACE on HotpotQA. Last-token targeting sharply reduces attribution coverage, from 0.384 to 0.161, and degrades both RISE and MAS. This confirms that a single token cannot capture the full attribution picture of a multi-token output, and validates the need for span-wise aggregation.

**Positional weighting.** One could also weight target positions non-uniformly, emphasizing the response tokens deemed most important. In our setting the target span comprises all response tokens after the reasoning chain, and there is no task-agnostic signal indicating which of them should receive more weight. Positional weighting would thus require task-specific knowledge that is unavailable in general, so we leave a principled instantiation of it as a direction for future work.

**Semantic cancellation.** Because aggregation sums token vectors, two positions carrying opposing semantics could in principle partially cancel before the proximity is computed. We did not observe this effect harming results in practice, since the L1-based proximity is evaluated on the aggregated target and our experiments show consistent gains across tasks, but we note it as a caveat that motivates the alternative operators discussed above.

*Table 10.* **Last-token targeting versus span-wise aggregation** on HotpotQA. Attributing only the final token sharply reduces coverage and faithfulness.

| Metric | FLASHTRACE | Last-Token |
|---|---|---|
| Attribution Coverage ↑ | **0.384** | 0.161 |
| RISE ↓ | **0.033** | 0.214 |
| MAS ↓ | **0.128** | 0.354 |

## M.2. Span Granularity

FLASHTRACE can construct the target span at either token-level or sentence-level granularity. The two are equivalent in cost: both perform a single aggregation pass, so there is no efficiency difference between them. The distinction is purely one of information resolution. Token-level spans preserve the finest-grained attribution signal, whereas sentence-level spans aggregate within-sentence detail away, yielding coarser but more stable attributions. All experiments in this paper operate at token-level granularity, which is the most demanding case, so sentence-level results should be at least as strong.

## N. Qualitative Failure Analysis

To delineate the boundary of applicability of FLASHTRACE, we summarize the conditions under which it succeeds and the failure modes we observe when those conditions are violated.

FLASHTRACE performs best when the evidence supporting an output is clear and localized. Multi-hop reasoning over specific named entities, such as the HotpotQA setting, is a favorable case: the relevant input tokens form a small, well-separated set, and span-wise aggregation concentrates attribution mass on them cleanly.

Two failure modes appear when this assumption breaks down. The first is *diffuse evidence*: when the information supporting an output is spread across many tokens that each contribute only a small partial signal, the aggregated target representation mixes these contributions, and the L1 proximity distributes mass thinly across the diffuse set rather than recovering any single token sharply. In this regime key evidence can fall outside the top-ranked tokens even though the overall attribution direction is correct. The second is *spurious salience of structural tokens*: high-frequency formatting or connective tokens that co-occur with the target span can absorb a share of attribution mass that exceeds their semantic role, because they participate in the residual stream that feeds the target.

Both modes reflect the efficiency-precision trade-off inherent to span-wise aggregation and L1 proximity. The same aggregation that makes FLASHTRACE a single-pass method also removes the per-token resolution needed to disentangle many small, similar contributions. Practitioners interpreting FLASHTRACE outputs on tasks with diffuse or highly nuanced token dependencies should therefore treat the resulting distributions as coarse localizations rather than precise token-level evidence, and may complement them with the exhaustive token-level rollout of Appendix D.3 when fine-grained precision is required.

