# OpenReview forum: "Towards Long-Horizon Interpretability: Efficient and Faithful Multi-Token Attribution for Reasoning LLMs"
_ICML.cc/2026/Conference — ICML 2026 spotlight_

### Official Review · Reviewer_nQ4N · 2026-02-14

**Soundness:** 4
**Presentation:** 4
**Significance:** 3
**Originality:** 4
**Overall Recommendation:** 5
**Confidence:** 3

**Summary:**

This paper studies long-horizon interpretability for reasoning LLMs, where standard token-level attribution overfocuses on intermediate reasoning tokens and misses the original inputs. It proposes FlashTrace, which computes multi-token attributions efficiently by aggregating attention and residual contributions over target spans, then optionally tracing influence recursively through the reasoning chain back to the input. Experiments on long-context retrieval tasks from RULER, multi-hop QA, MATH, and code generation show improved faithfulness and large speed and memory gains.

**Compliance With Llm Reviewing Policy:**

Affirmed.

**Final Justification:**

The paper makes a solid and original contribution to long-horizon attribution for reasoning LLMs, with clear practical gains in faithfulness, efficiency, and scalability. My main concerns were about evaluation robustness and long-context validation, and the rebuttal addressed them well with additional experiments and clarifications. I therefore maintain my Accept recommendation.

**Key Questions For Authors:**

1. How sensitive are results to the span construction and aggregation granularity, such as token spans versus sentence spans?
2. Can you report recovery and faithfulness trends on RULER as context grows into the multi-thousand token regime, not only efficiency scaling?
3. What happens when the model answer is wrong or the chain-of-thought is inconsistent, since the current eval filters to correct outputs?

**Limitations:**

The paper would benefit from a direct discussion of method limits, such as approximation error from span aggregation and normalization linearization, plus dependence on access to attention internals at scale.

**Strengths And Weaknesses:**

Overall, this is a solid and useful contribution that targets a real bottleneck in attribution for long-form LLM generations.

Strengths:
- Clear problem framing with an empirical motivation showing how reasoning tokens absorb attribution mass and hurt evidence recovery.
- Span-wise aggregation gives strong practical wins, with large speedups and better scaling in memory for long target spans.
- Faithfulness gains are consistent across several tasks, and the method stays stable across reasoning lengths.
- The study includes helpful analysis, including comparison to an exhaustive rollout baseline and a hop-depth ablation. It also has good presentation figures and tables.

Weaknesses:
- Faithfulness eval relies on a modified perturbation schedule and filters to only correct generations, so it is unclear how robust conclusions are under alternative settings.
- Most RULER faithfulness results use a fixed 1024-token context, which weakens the long-context claim for attribution quality beyond that range.
- Practical deployment may still be limited by needing attention maps across layers, so more guidance on compute and memory tradeoffs would help.

---

> ### Author Rebuttal · Authors · 2026-03-28
>
> Thank you for the constructive review. We appreciate that the reviewer found our contribution solid and useful. The feedback has motivated us to run additional experiments that we believe strengthen the paper. We address each point below.
>
> > W1 & Q3: Faithfulness evaluation relies on a modified perturbation schedule and filters to only correct generations, so it is unclear how robust conclusions are under alternative settings. What happens when the model answer is wrong or the chain-of-thought is inconsistent?
>
> Thank you for raising this point.
>
> - About perturbation schedule: our schedule (Appendix E.2) monotonically increases the masking ratio, preserving the properties that make RISE/MAS valid. The only change is using fixed step-size intervals to reduce forward passes without altering ranking semantics. Additionally, the attribution coverage ratio is consistent with the MAS/RISE score, supporting our claims.
> - Regarding correctness filtering: flashtrace itself does not depend on model correctness, as it traces information flow regardless of whether the model's answer is right or wrong. The correctness filter was applied because the Recovery Rate metric requires correctness to measure whether the attribution recovers the ground-truth evidence. RISE/MAS, on the other hand, do not require correctness. Motivated by this concern, we ran additional experiments on HotpotQA with no correctness filtering, using only RISE/MAS. flashtrace consistently outperforms baselines under this unfiltered setting:
>
> | Context Length | Method | RISE ↓ | MAS ↓ | Time (s) |
> |---|---|---|---|---|
> | 2048 | flashtrace (ft-ifr) | **0.0361** | **0.1175** | 3.58 |
> | 2048 | IFR | 0.0586 | 0.1386 | 5.10 |
> | 2048 | AttnLRP | 0.3832 | 0.5436 | 1.19 |
> | 4096 | flashtrace (ft-ifr) | **0.0250** | **0.1063** | 9.12 |
> | 4096 | IFR | 0.0398 | 0.1150 | 12.71 |
> | 4096 | AttnLRP | 0.2316 | 0.2908 | 2.63 |
>
>
> > W2 & Q2: Most RULER faithfulness results use a fixed 1024-token context, which weakens the long-context claim for attribution quality beyond that range.
>
> We agree that this deserved further evaluation. Motivated by your suggestion, we ran additional experiments on HotpotQA on 2048 and 4096 length (also in above table): flashtrace achieves RISE 0.036 at 2048 tokens and 0.025 at 4096 tokens, showing that faithfulness actually improves at longer contexts while maintaining consistent advantage over all baselines. We will include these results in the revision.
>
> > W3: Practical deployment may still be limited by needing attention maps across layers, so more guidance on compute and memory tradeoffs would help.
>
> Thank you for this valuable suggestion. In practice, we can recompute attention map on-the-fly in flashtrace instead of caching the whole map to reduce memory usage. We benchmarked this mode on Qwen3-4B (bf16, 2048 tokens, 5 runs):
>
> | Mode | Time (s) | Peak GPU Memory (GiB) |
> |---|---|---|
> | Cached Attention Maps | 1.98 ± 0.27 | 35.43 ± 0.00 |
> | On-the-fly Recomputation | 2.81 ± 0.06 | 13.43 ± 0.00 |
>
> On-the-fly recomputation trades 42% runtime for 62% memory savings, making flashtrace more viable on consumer GPUs. Users can select the mode that best fits their hardware. We will add this guidance to the revision.
>
> > Q1: How sensitive are results to the span construction and aggregation granularity, such as token spans versus sentence spans?
>
> In flashtrace, token-level and sentence-level spans both perform a single aggregation pass, so there is no efficiency difference. The distinction is purely in information resolution: token-level spans preserve the finest-grained attribution signal, while sentence-level spans would in principle lose within-sentence detail, yielding coarser but more stable attributions. Since all our experiments operate at token granularity (the hardest case), sentence-level results should be at least as strong. We will add this discussion in the revision.
>
> We hope these clarifications and additional experiments address the reviewer's concerns. We believe the reviewer's suggestions will make our paper stronger.

---

> > ### Author Rebuttal · Reviewer_nQ4N · 2026-04-01
> >
> > I thank the authors for their solid response. After reading it, I believe my questions have been mostly resolved, and the weaknesses of the paper are relatively minor compared with its contributions. At the same time, I believe the experiments demonstrate sufficient strengths of the method to justify acceptance. I would encourage the authors to include this additional analysis in the final version of the paper. I will therefore keep my rating unchanged as Accept and maintain my assessment that this is a technically solid paper worthy of acceptance.

---

> > > ### Author Response · Authors · 2026-04-08
> > >
> > > Thank you for the encouraging feedback and for the suggestions that motivated our additional experiments. We will incorporate the unfiltered faithfulness results, longer-context evaluations, and compute/memory guidance into the final version.

---

### Official Review · Reviewer_nuiV · 2026-02-15

**Soundness:** 4
**Presentation:** 4
**Significance:** 4
**Originality:** 3
**Overall Recommendation:** 5
**Confidence:** 4

**Summary:**

The paper introduces FLASHTRACE, an efficient multi-token attribution method that employs span-wise aggregation to compute attribution over multi-token targets in a single pass, while maintaining faithfulness. It formalize the multi-torkn attribution problems and target two umportant problems: efficiency bottleneck and faithfulness degradation since intermediate reasoning tokens absorb most of attributions. Extensive experiments and analysis showcases the great advantages of the proposed method.

**Compliance With Llm Reviewing Policy:**

Affirmed.

**Key Questions For Authors:**

1. I am a little confused about recursive attribution hop, i.e., when you get w_I^0 and w_T^0, how to delve into next step? what is new target span in the reasoning tokens?

**Limitations:**

Yes

**Strengths And Weaknesses:**

Strengths:

1. the problem is important and valuable.
2. the porposed method is simple-yet-effective, efficient and straightforward, directly focus on the identified challenges.
3. the writing and presentation is good, with several informative figures and tables.
4. the experiments and analysis are comprehensive and solid, with lots of persuasive results and settings.

Weakness

1. there is no significant weakness.

---

> ### Author Rebuttal · Authors · 2026-03-28
>
> Thank you for the positive evaluation and for recognizing the importance of the problem and the comprehensiveness of our experiments! We answer your questions as follows:
>
> > Q1: Recursive attribution hop — when you get $w_I^{(0)}$ and $w_T^{(0)}$, how to delve into the next step? What is the new target span in the reasoning tokens?
>
> Thank you for this question. Concretely, the new target span for Hop 1 is the **entire** reasoning token sequence $T$, where each token is weighted by its importance score $w_j^{(0)}$ from the previous hop. The aggregated target representation and source contribution then become (Section 4.3, Eq. 9–10):
>
> $$Y^{(1)} = \sum_{j \in T} w_j^{(0)} \cdot y_j, \quad Z^{(1)} = \sum_{j \in T} w_j^{(0)} \cdot z_{k \to j}$$
>
> The key intuition is that each recursive hop asks a progressively deeper causal question:
>  - Hop 0 identifies which reasoning tokens matter most for the **output**;
>  - Hop 1 then asks which earlier tokens drove **those** high-importance reasoning tokens.
>
> Tokens that received high importance in Hop 0 dominate this weighted target, while low-importance tokens contribute minimally. We then apply the same proximity-based importance calculation to this weighted target, yielding a new distribution $w^{(1)}$ that traces importance one step further back along the reasoning chain. Figure 3 visualizes this process: Hop 1 concentrates on reasoning tokens nearest to the output, while Hop 2 shifts attribution toward earlier reasoning steps and the original input.
>
> We will add a brief clarifying note in the revised paper to make the transition between hops clearer.

---

> > ### Author Rebuttal · Reviewer_nuiV · 2026-04-05
> >
> > Thanks the authors for the clarification, I decide to maintain my score.

---

> > > ### Author Response · Authors · 2026-04-08
> > >
> > > Thank you for the positive evaluation. We will add a clarifying note on the recursive hop transition to make the mechanism easier to follow.

---

### Official Review · Reviewer_Rt4D · 2026-03-09

**Soundness:** 3
**Presentation:** 3
**Significance:** 3
**Originality:** 3
**Overall Recommendation:** 4
**Confidence:** 3

**Summary:**

The paper addresses two critical limitations of current token-level attribution methods when applied to reasoning LLMs: the $O(M \cdot N)$ efficiency bottleneck in multi-token attribution and "faithfulness drop" caused by intermediate reasoning tokens absorbing attribution mass. The authors propose FLASHTRACE, which combines span-wise aggregation for $O(N)$ complexity per pass and a recursive attribution mechanism to trace importance from outputs, through reasoning chains, back to original inputs.

**Compliance With Llm Reviewing Policy:**

Affirmed.

**Final Justification:**

The authors have addressed my concerns.

**Key Questions For Authors:**

Q1. Linearization Error: How does the fidelity of the element-wise scaling approximation for normalization change as the model depth or context length increases?

Q2. Adaptive Hop Selection: Could you implement a stopping criterion for the recursive process rather than using a fixed $K$?

**Limitations:**

L1. The method requires access to internal model states (activations and attention maps), making it inapplicable to closed-source API-based models.

L2. This work is not expected to have any direct negative social consequences.

**Strengths And Weaknesses:**

S1. Rigorous Empirical Evaluation: The authors evaluate FLASHTRACE across diverse tasks, including long-context retrieval (RULER), mathematical reasoning (MATH), multi-hop QA (HotpotQA), and code generation (Aider).

S2. Significant Efficiency Gains: By pre-aggregating scalar attention weights, the method reduces complexity from $O(M \cdot N \cdot D)$ to $O(N \cdot D)$, achieving over 130x speedup.

S3. Clarity of Presentation: The paper is well-structured and uses effective visualizations to illustrate complex recursive information flows.

W1. Heuristic Linearization Risks: The method approximates non-linear normalization layers (RMSNorm/LayerNorm) as element-wise scaling. While practical, the paper lacks a quantitative analysis of how this approximation error accumulates across very deep reasoning chains.

W2. Lack of Adaptive Recursion: Most evaluations are conducted with a fixed $K=1$. While Appendix H provides a sensitivity analysis for $K$, the method lacks a dynamic mechanism to determine the optimal number of hops based on the complexity of the reasoning chain.

---

> ### Author Rebuttal · Authors · 2026-03-28
>
> Thank you for the constructive feedback. We are encouraged that the reviewer recognizes the rigor of our evaluation and efficiency gains. We address each concern below.
>
> > W1 & Q1: The method approximates non-linear normalization layers as element-wise scaling. The paper lacks a quantitative analysis of how this approximation error accumulates across very deep reasoning chains. How does the fidelity of this approximation change as model depth or context length increases?
>
> This is an important consideration for attribution methods. The element-wise scaling approximation for normalization layers is an established technique from the ALTI/IFR framework (Ferrando et al., 2022; Ferrando & Voita, 2024), which treats normalization scaling factors as locally constant. FlashTrace inherits this approach without modification (Section 4.1, L208–211).
>
> That said, we do validate the end-to-end approximation quality. Table 3 shows FlashTrace closely matches Exhaustive Token-Level Rollout in faithfulness (RISE: 0.128 vs. 0.116; MAS: 0.205 vs. 0.193) with much shorter runtime. We will add a more explicit discussion of the linearization's provenance in the revision.
>
> > W2 & Q2: Most evaluations use a fixed K=1. While Appendix H provides a sensitivity analysis, the method lacks a dynamic mechanism to determine the optimal number of hops based on the complexity of the reasoning chain. Could you implement a stopping criterion rather than using a fixed K?
>
> Thank you for this thoughtful suggestion. Our analysis in Appendix H (L1131–1143, Table 7, Figure 7) examines the effect of recursion depth K. Table 7 reports faithfulness metrics on MorehopQA with varying K:
>
> | Configuration | RISE ↓ | MAS ↓ |
> |---|---|---|
> | No Recursion (K=0) | 0.127 | 0.209 |
> | 1 Hop (K=1) | 0.128 | 0.205 |
> | 2 Hops | 0.130 | 0.206 |
> | 3 Hops | 0.133 | 0.209 |
>
> K=1 achieves the best faithfulness, with diminishing returns beyond due to noise accumulation. Figure 7 additionally confirms that the majority of attribution mass redistribution occurs within the first hop. In principle, a stopping criterion could be based on the fraction of attribution mass redistributed at each hop, which drops sharply after K=1. However, since K=1 consistently performs best across settings, a fixed K=1 is a simple and empirically robust default. We acknowledge that designing a more principled adaptive mechanism is an interesting direction for future work, and we will include this discussion in the revised paper. We thank the reviewer again for the helpful comments. We believe this feedback will make our paper stronger.

---

> > ### Author Rebuttal · Reviewer_Rt4D · 2026-04-03
> >
> > Thank you for the clarification.  I will maintain my score.

---

> > > ### Author Response · Authors · 2026-04-08
> > >
> > > Thank you for the thoughtful review and for confirming that your concerns have been addressed. We will include the linearization provenance discussion and the adaptive recursion analysis in the revised paper.

---

### Official Review · Reviewer_s37z · 2026-03-13

**Soundness:** 3
**Presentation:** 3
**Significance:** 2
**Originality:** 3
**Overall Recommendation:** 4
**Confidence:** 3

**Summary:**

This paper introduces FLASHTRACE, an attribution method designed for long-horizon interpretability of reasoning LLMs, where the target to explain is a multi-token span. FLASHTRACE tackles two practical issues, reducing the computational burden of attributing long target spans token by token and mitigating faithfulness degradation when attribution mass concentrates on intermediate reasoning tokens rather than tracing back to the original inputs. FLASHTRACE combines span-wise aggregation, which attributes an entire target span in a single pass, with recursive attribution that propagates importance through intermediate reasoning chains.

**Compliance With Llm Reviewing Policy:**

Affirmed.

**Key Questions For Authors:**

1. In the experiments, do the authors cache all attention maps across layers, or recompute them on-the-fly? What are the concrete time vs. memory trade-offs under both strategies?
2. The authors use vector summation for aggregating target spans. Could this be misleading when span tokens have strong order dependence or semantic cancellation? Did the authors evaluate alternatives (mean pooling, positional weighting, last-token) via ablations?

**Limitations:**

Yes.

**Strengths And Weaknesses:**

**Strengths**：
1. The paper studies a practically important interpretability problem for modern reasoning/agentic LLMs, and the proposed setting is well motivated.
2. This paper is presented in a clear and well-structured way, making the motivation, algorithmic design, and complexity implications straightforward to understand.
3. The paper provides extensive experiments and scaling results that substantiate the efficiency benefits and overall effectiveness of the approach.

**Weaknesses**：
1. The paper focuses on token-level attribution baselines. It would benefit from discussing other interpretability methods such as transcoders[1] and circuit-tracer[2].
2. The scoring mechanism relies on proximity (L1-based) measures and linearization approximations. Although this choice is computationally efficient, it weakens the causal guarantees of the attribution scores, so a high score should not be interpreted as definitive causal influence. It would therefore be helpful for the paper to more clearly separate correlation-based evidence from stronger causal claims.
3. This paper would benefit from qualitative examples where the method fails (e.g., key evidence not recovered, spurious tokens highlighted), clarifying the boundary of applicability.
References:
[1] Dunefsky J, Chlenski P, Nanda N. Transcoders find interpretable llm feature circuits[J]. Advances in Neural Information Processing Systems, 2024.
[2] Hanna M, Piotrowski M, Lindsey J, et al. Circuit-tracer: A new library for finding feature circuits[C]//Proceedings of the 8th BlackboxNLP Workshop: Analyzing and Interpreting Neural Networks for NLP. 2025.

---

> ### Author Rebuttal · Authors · 2026-03-28
>
> Thank you for the constructive feedback and for recognizing the practical importance of the problem, the clarity of presentation, and the extensiveness of our experiments. We address each concern below.
>
> > W1: The paper focuses on token-level attribution baselines. It would benefit from discussing other interpretability methods such as transcoders and circuit-tracer.
>
> Thank you for highlighting these references. Transcoders decompose MLP layers into interpretable feature dictionaries and circuit-tracer identifies minimal circuits responsible for specific behaviors. Both operate at the feature-circuit level, identifying **which internal components** drive a behavior.
>
> Flashtrace operates at a different level: based on information flow analysis (Ferrando et al., 2022; Ferrando & Voita, 2024), it quantifies how much information from source tokens reaches a target span's representation. This measures **relevance** (which tokens inform output) instead of circuit-level **causality**. These paradigms are complementary: one may use flashtrace to quickly identify key input tokens, then apply circuit-tracing to those regions. We will add this important discussion to Related Work in the revision.
>
> > W2: The scoring mechanism relies on proximity (L1-based) measures and linearization approximations. It would be helpful to more clearly separate correlation-based evidence from stronger causal claims.
>
> Thank you for this suggestion. We will explicitly frame flashtrace as measuring **informational contribution** rather than **causal influence** in the revision. That said, our intervention-based faithfulness evaluation validates that these relevance scores indirectly predict causal importance.
>
> > W3: Qualitative examples where the method fails would clarify the boundary of applicability.
>
> Thank you for the valuable suggestion. We will include a qualitative failure analysis in the revision. From our experiments, flashtrace performs best when evidence signals are clear and localized (e.g., multi-hop reasoning over specific named entities like HotpotQA), while it degrades for fine-grained, nuanced token dependencies where multiple tokens contribute similar partial information. This reflects the efficiency–precision trade-off of span-wise aggregation and L1 proximity. We will add concrete failure examples with discussion in the Appendix.
>
> > Q1: Do the authors cache attention maps across all layers, or recompute them on the fly? What are the concrete time vs. memory trade-offs?
>
> Great question. We cache attention maps of all layers during forward passing in our implementation. In practice, one can recompute them on the fly (as flashtrace computes in the same order as the forward pass), trading memory $O(L \cdot H \cdot |\mathbf{S}|^2)$ vs. $O(H \cdot |\mathbf{S}|^2)$ for latency. We benchmark both modes on Qwen3-4B (bf16), 2048 tokens, 5 runs:
>
> | Mode | Time (s) | Peak GPU Memory (GiB) |
> |---|---|---|
> | Cached | 1.98 ± 0.27 | 35.43 ± 0.00 |
> | On-the-fly | 2.81 ± 0.06 | 13.43 ± 0.00 |
>
> On-the-fly mode trades 42% more latency for 62% memory reduction, making it useful for resource-constrained settings. We will add this analysis to the revision.
>
> > Q2: Could vector summation for aggregating target spans cause semantic cancellation? Did the authors evaluate alternatives (mean pooling, positional weighting, last-token)?
>
> Thank you for this thoughtful question. In principle, semantic cancellation could occur when two positions carry mutually canceling semantics. We evaluate the alternatives below:
>
> - **Mean pooling:** The IFR baseline in our paper is essentially the single-token fallback of flashtrace, adapted with mean pooling over individual token attributions for multi-token targets. As shown in Table 1, IFR shows lower recovery rate but better faithfulness in some long-context retrieval settings. However, mean pooling moves aggregation after the L1 proximity operation, breaking the linearity needed for efficient pre-aggregation (Section 4.2), making it substantially slower.
>
> - **Last-token:** We ran an experiment targeting only the last token on HotpotQA:
>
> | Metric | flashtrace | Last Token |
> |---|---|---|
> | Attribution Coverage | 0.384 | 0.161 |
> | Faithfulness (RISE / MAS) | 0.033 / 0.128 | 0.214 / 0.354 |
>
> Last-token attribution sharply reduces coverage (0.161 vs. 0.384), confirming that a single token cannot capture the full attribution picture for multi-token output and validating the need for span-wise aggregation.
>
> - **Positional weighting:** Our target span comprises all response tokens after reasoning. Positional weighting would require task-specific knowledge of which response tokens are more important, which is unavailable in our setting. We note this as a promising future direction in the revision.
>
> We thank the reviewer again for the insightful comments. We believe these suggestions have helped us strengthen both the paper and the evaluation, and we will incorporate all the discussed improvements in the revision.

---

> > ### Author Rebuttal · Reviewer_s37z · 2026-04-03
> >
> > Thanks the authors for the clarification and additional experiments.

---

> > > ### Author Response · Authors · 2026-04-08
> > >
> > > We appreciate the reviewer's time in reading our rebuttal. We will incorporate all discussed improvements into the camera-ready version, including the discussion of transcoders/circuit-tracer, clearer framing of correlation vs. causation, and qualitative failure examples.

---

### Decision · Program_Chairs · 2026-04-30

**Decision:**

Accept (spotlight)

**Comment:**

This paper presents Flash Trace, an efficient multi-token attribution method designed for long-horizon interpretability in reasoning LLMs by employing span-wise aggregation and a recursive attribution mechanism. The reviewers agreed that the work tackles a practically important problem, praising its clear presentation, rigorous empirical evaluation across diverse tasks, and the significant computational speedups achieved. While reviewers initially raised some concerns regarding evaluation robustness, context length limitations, and approximation risks for normalization layers, the authors provided comprehensive responses during the rebuttal period. Their response included additional unfiltered faithfulness evaluations, experiments on extended context lengths up to 4096 tokens, and detailed memory versus runtime trade-off analyses. Given the paper's solid algorithmic design, substantial efficiency gains, and the authors' thorough incorporation of reviewer feedback, I recommend accepting this paper for publication at ICML 2026.